# Functional compensation precedes recovery of tissue mass following acute liver injury

Chad M. Walesky [1,12], Kellie E. Kolb [2,3,4,12], Carolyn L. Winston [1], Jake Henderson [1], Benjamin Kruft [1], Ira Fleming [2,3,4], Sungjin Ko [5], Satdarshan P. Monga [5], Florian Mueller [6], Udayan Apte [7], Alex K. Shalek [2,3,4,8,13 ✉] & Wolfram Goessling [1,3,8,9,10,11,13 ✉]

The liver plays a central role in metabolism, protein synthesis and detoxification. It possesses unique regenerative capacity upon injury. While many factors regulating cellular proliferation during liver repair have been identified, the mechanisms by which the injured liver maintains vital functions prior to tissue recovery are unknown. Here, we identify a new phase of functional compensation following acute liver injury that occurs prior to cellular proliferation. By coupling single-cell RNA-seq with in situ transcriptional analyses in two independent murine liver injury models, we discover adaptive reprogramming to ensure expression of both injury response and core liver function genes dependent on macrophage-derived WNT/β-catenin signaling. Interestingly, transcriptional compensation is most prominent in non-proliferating cells, clearly delineating two temporally distinct phases of liver recovery. Overall, our work describes a mechanism by which the liver maintains essential physiological functions prior to cellular reconstitution and characterizes macrophage-derived WNT signals required for this compensation.

[1] Genetics Division, Brigham and Women's Hospital, Harvard Medical School, Boston, MA 02115, USA. [2] Institute of Medical Engineering & Science (IMES), Department of Chemistry, and Koch Institute for Integrative Cancer Research, Massachusetts Institute of Technology, Cambridge, MA 02139, USA. [3] Broad Institute of MIT and Harvard, Cambridge, MA 02142, USA. [4] Ragon Institute of MGH, MIT and Harvard, Cambridge, MA 02139, USA. [5] Department of Pathology, University of Pittsburgh, School of Medicine; and Pittsburgh Liver Research Center, University of Pittsburgh and University of Pittsburgh Medical Center, Pittsburgh, PA 15261, USA. [6] Imaging and Modeling Unit, Institut Pasteur, UMR 3691CNRS, C3BI USR 3756 IP CNRS, Paris, France. [7] Department of Pharmacology, Toxicology, and Therapeutics, University of Kansas Medical Center, Kansas City, KS 66160, USA. [8] Harvard-MIT Division of Health Sciences and Technology, Boston, MA 02115, USA. [9] Dana-Farber Cancer Institute, Boston, MA 02215, USA. [10] Harvard Stem Cell Institute, Cambridge, MA 02134, USA. [11] Division of Gastroenterology, Massachusetts General Hospital, Harvard Medical School, Boston, MA 02114, USA. [12] These authors contributed equally: Chad M. Walesky, Kellie E. Kolb. [13] These authors jointly supervised this work: Alex K. Shalek, Wolfram Goessling. ✉email: shalek@mit.edu; wgoessling@partners.org

The liver is a vital organ with a wide array of functions, including homeostasis of glucose, protein, and lipid metabolism; production of bile; synthesis of critical serum proteins; and metabolism of endogenous and xenobiotic toxins[1]. The liver experiences frequent toxic insults that can result in cellular injury and death. Acetaminophen (APAP) overdose is the most common cause of toxic liver injury in the United States[2]. Another cause for acutely reduced liver function is surgical resection, often used to remove liver tumors or cancer metastases[3]. Typically, any injury to the liver is followed by liver regeneration through cellular proliferation[4]. While the process and signals involved in liver regeneration have been studied in great detail, it remains unclear how an organism can survive the period immediately following the loss of substantial functional liver tissue.

Hepatocytes are the major parenchymal cells of the liver and are organized into lobules with coordinated blood flow from the portal to the central vein. Within each lobule, hepatocyte gene expression is heterogenously structured, governed by metabolic demands and signaling pathways, supporting distinct hepatocellular functionality across the periportal, midlobular, and pericentral zones[1]. The impact of either zone-specific injury or global loss of liver mass on gene expression in these different regions has not been examined.

Pioneering studies on liver regeneration have primarily focused on the signaling mechanisms that underlie re-establishment of lost cell mass through hepatocyte proliferation and coordinated angiogenesis[4–7]. While cell cycle initiation can be detected at 12–24 h after injury, cell proliferation typically begins only after ~24–30 h and does not peak until ~40–48 h[7,8]. This is in great contrast to regenerative responses in other organs, such as skin, where cell proliferation is initiated within a few hours following injury[8]. The reason for this delay in hepatocyte proliferation has been enigmatic.

Here, we investigate compensatory responses in the liver during key phases of liver regeneration in both a toxic (APAP) injury model and following surgical resection (partial hepatectomy, PH) (Fig. 1a). Utilizing the powerful combinatorial approach of Seq-Well, a massively-parallel single-cell RNA-seq (scRNA-Seq) platform ideally suited for fragile cells like hepatocytes, and single-molecule fluorescent in situ hybridization (smFISH), we define transcriptional changes after injury[9]. We discover that remaining hepatocytes functionally compensate for lost liver mass by increased transcriptional output of key hepatocyte genes. Importantly, hepatocytes also alter their zone-specific functional identities within the liver lobule to maintain global expression of select transcripts. We find that hepatocyte functional compensation precedes the peak phase of cellular proliferation and that cycling cells do not participate in this process to the same degree as non-cycling hepatocytes during the regeneration phase. Both cycling and non-cycling cells show upregulation of Wnt signaling targets—known to play a central role in normal hepatocyte development, maintenance, and liver regeneration;[10–24] we demonstrate that compensation depends on intact Wnt/β-catenin activation through macrophage-secreted Wnts. Overall, our results identify previously unappreciated plasticity among hepatocytes during a newly discovered compensatory phase after liver injury, as well as Wnt/β-catenin signaling as a therapeutically relevant pathway for maintaining and re-establishing homeostatic liver function.

## Results

**Transcriptional adaption after liver injury.** To assess global transcriptional shifts in hepatocytes at single-cell resolution following acute liver injury, we employed scRNA-Seq to characterize response dynamics in both PH and APAP models, capturing the injury, regeneration, and termination phases of liver regeneration[4] (Fig. 1b, c). We profiled a total of 16,019 cells across 19 different experiments to an average sequencing depth of >48,000 reads/cell (Supplementary Fig. 1a–c, Supplementary Methods). Immune and endothelial cell types, as well as low-quality cells, were filtered out from the dataset, retaining 10,762 high-quality hepatocyte transcriptomes for subsequent analyses (Supplementary Fig. 1d, e, Supplementary Data 1, Methods). Shared nearest neighbor clustering (SNN) visualized on a t-Stochastic Neighbor Embedding (t-SNE) plot revealed hepatocyte populations that cluster by injury model and post-injury time point (Fig. 1d, Methods). While hepatocytes from each untreated mouse clustered independently, the injury samples grouped by time point and injury type, rather than mouse of origin, indicating that the transcriptional response to injury causes individual hepatocytes to become more similar to one another. To confirm that this clustering captures biological, rather than technical, variation, we performed differential expression to identify genes unique to each cluster. Clusters were defined by many genes related to liver function, injury response, and oxidative stress (Fig. 1e, Supplementary Data 3), and technical gradients led to variation within, rather than across, clusters (nGene, nUMI; Supplementary Fig. 2). Regression over technical variables (i.e., number of genes) largely removed these technical gradients, but preserved other, biologically important signals; removal of PC1, which captured technical effects, similarly resulted in a reduction of technical signals while preserving key biological ones. Since regression changed very little, other than downweighting technical differences in cell quality, and the biological signals on which this work focuses were robust to regression, we opted to use the non-regressed dataset in our downstream analysis to avoid possible introduction of artificial variation.

APAP injury resulted in pericentral necrosis after 6 h as demonstrated by histological analysis (hereafter A6; Fig. 1b, c). Hepatocytes scoring high for a pericentral hepatocyte signature (PCHSig) were absent at 6 h post-APAP (A6, Fig. 1f). Surprisingly, at 24 h post-APAP, the pericentral hepatocyte expression signature returned (A24, Fig. 1f), despite histology showing persistent pericentral necrosis (A24, Fig. 1b, c). In particular, expression of two typically pericentrally restricted genes—Cyp2e1, responsible for metabolizing APAP, and Glul, which assimilates ammonia into glutamine—was maintained, or returned, following pericentral injury. For example, Cyp2e1 + hepatocytes decreased from 67% (Untreated, UT) to 5% (A6), but returned back to 46% by 24 h with no significant change in the number of Glul + hepatocytes at any time point. These results suggest the intriguing possibility of compensatory expression of pericentral genes by non-pericentral hepatocytes.

scRNA-seq is a powerful tool for determining global transcriptional changes in subsets of cells from a heterogenous organ, such as the liver; however, meticulous validation is necessary to overcome some limitations of this rapidly-evolving technology, such as inefficiencies in transcript capture. To validate these findings in a spatiotemporal context, we assessed the distribution of the pericentral markers Cyp2e1 and Glul using highly sensitive smFISH analysis (Fig. 2a–e; Supplementary Figs. 3, 4). Cyp2e1 extended further into the lobular midzone following APAP exposure, with pericentral necrosis at A6 and A24 (Fig. 2b, Supplementary Fig. 4). Expression then normalized at A48, following the cell proliferative response. Glul expression is normally restricted to a single layer of cells surrounding the central vein[25], which underwent necrosis following APAP overdose (Fig. 2b). Surprisingly, we observed that Glul was now expressed at low levels across the entire liver lobule (A6, A24, Fig. 2b; Supplementary Fig. 4). Glul expression patterns return to

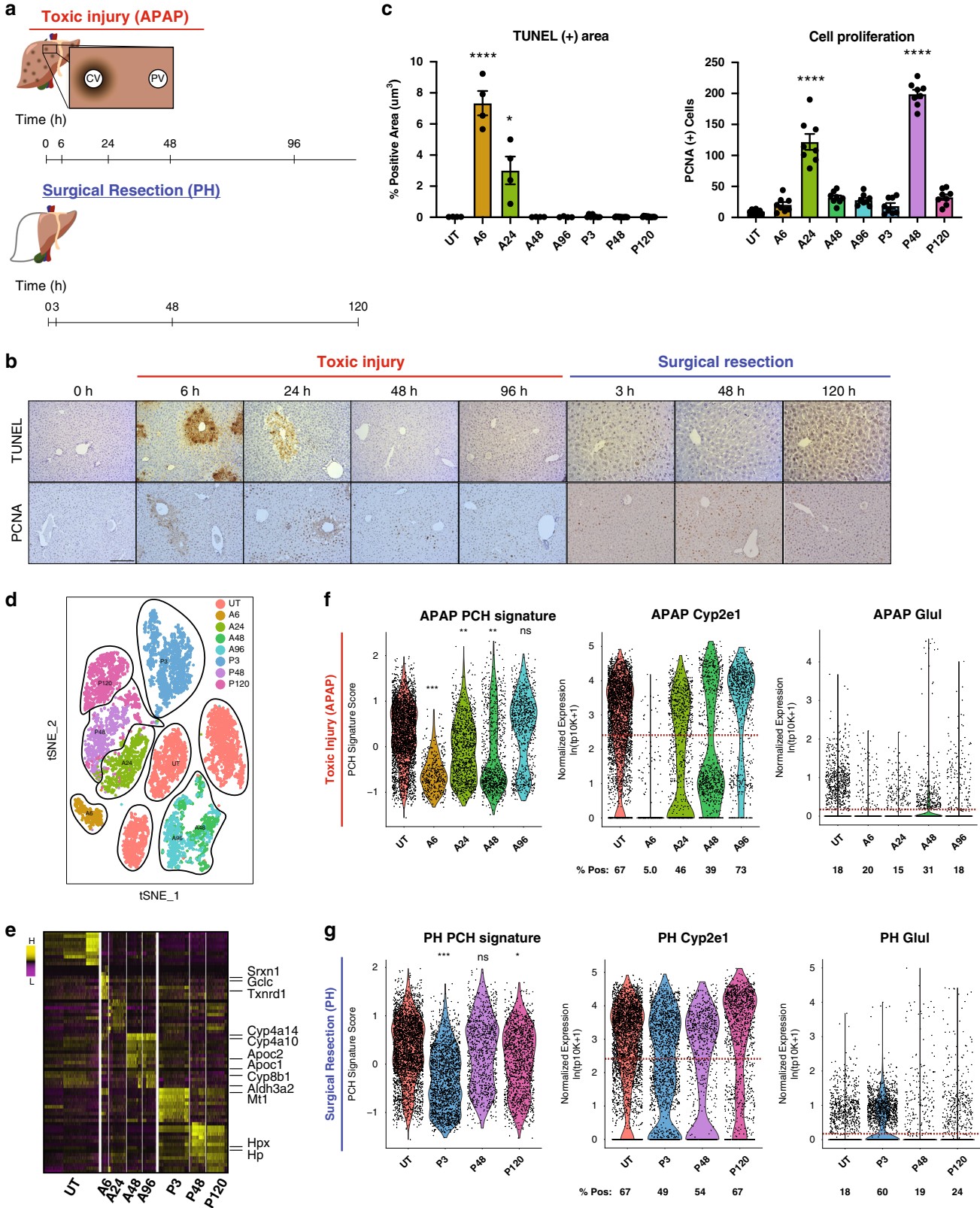

normal by A48 (Fig. 2b; Supplementary Fig. 4). These findings reveal compensatory expression of pericentral genes through adaptive reprogramming of midzonal hepatocytes after pericentral toxic injury.

In contrast to APAP, PH does not result in zone-dependent injury but represents a massive loss of ~70% of liver cell mass (compared to ~10% total cell loss after APAP injury), imposing extreme functional demand on remaining hepatocytes. Similarly to APAP toxicity, functional compensation was also observed after PH, evident from a dramatic increase in *Glul* + hepatoctyes (Fig. 1g) from 18% (Control) to 60% (P3). This is further supported by the observation that *Cyp2e1* + hepatocytes only

**Fig. 1 Hepatocyte response to toxic and surgical liver injuries. a** Time course depicting analysis time points during liver injury recovery following APAP overdose or PH. **b** Murine liver sections (5 μm, $n = 3$) show necrotic TUNEL-positive (brown, top) and proliferative PCNA-positive (brown, bottom) cells. Scale bar is 100 μm. **c** Bar graphs quantifying total TUNEL- and PCNA-positive area. Error bars are s.e.m., $P < 0.05$ (*), < 0.01 (**), and < 0.0001 (****) calculated using unpaired $t$ test with Welch's correction (two-tailed). **d** t-SNE plot of all high-quality hepatocytes (Methods) in the scRNA-Seq dataset. Cells are colored by injury mode and time point. SNN clusters outlined in black. **e** Heatmap of marker genes for all clusters outlined in (**d**). **f, g** Pericentral Hepatocyte Signature Score (PCH Signature Score) (left). Violin plot of normalized expression of *Cyp2e1* (middle) and *Glul* (right); percent positive calculated as percentage of total cells in each condition above average normalized genes expression (dashed red line). Untreated (UT) and each post-treatment are plotted for APAP (**f**) and PH (**g**). Source data provided as a Source Data file.

decrease by 18% (67 to 49%) at P3 (Fig. 1g). smFISH analysis confirmed expanded expression zones and total expression levels for both *Cyp2e1* and *Glul* in PH (Fig. 2c, e; Supplementary Fig. 4a, b). In addition, we observed functional compensatory upregulation of the periportal marker *Arg1*, suggesting that adaptive reprogramming of hepatocytes is not exclusive to pericentral genes (Supplementary Fig. 4c, d). These results highlight that hepatocytes alter their transcriptional output to maintain the expression of zonally expressed genes otherwise lost due to injury.

**Liver injury causes both injury- and non-specific responses.** To further define shared and unique responses in both liver injury models, we determined differentially expressed genes (DEGs) between each treatment condition and untreated controls (UT), and then pooled results to reveal composite DEG results for APAP and PH (Methods, Supplementary Data 4), for which we performed specific validations using smFISH (Fig. 3, Supplementary Figs. 6–10). A number of gene expression alterations could be attributed to injury-specific effects. A large number of gene expression changes, however, were shared between the two models, indicating that they do not reflect the specific nature of injury (Fig. 3a, Supplementary Fig. 5, Supplementary Data 5). Gene set analysis (GSA) revealed an enrichment of pathways involved in toxic injury within the APAP model (Supplementary Fig. 5, Supplementary Data 6)[26]. The PH model, meanwhile, exhibited an enrichment among pathways involved in cell proliferation, possibly due to differences in the extent of injury between the two models (Supplementary Fig. 5, Supplementary Data 6). Following APAP overdose, there was a substantial antioxidant response by GSA and expression of individual antioxidant response genes, such as *thioredoxin reductase* (*Txnrd1*) and *glutamate-cysteine ligase subunit c* (*Gclc*) (Fig. 3b). Surprisingly, smFISH analysis revealed an upregulation of *Txnrd1* and *Gclc* across the entire liver lobule reaching into the periportal region (Fig. 3b, Supplementary Figs. 6–10). These findings suggest that some injury responses are not exclusive to lobular localization of injury.

DEG analysis further highlighted the breadth of the functional compensatory response with a large number of the most significantly upregulated genes being involved in classic hepatic functions, such as secreted protein synthesis, gluconeogenesis, blood clotting, and ion homeostasis (Fig. 3, Supplementary Figs. 6–10). Albumin is the most abundant serum protein and is produced by all hepatocytes across the liver lobule, with the highest expression in the periportal region. Acute injury in both models resulted in a dramatic upregulation of albumin across the entire liver lobule beginning at the earliest observed time points (A6 and P3) (Fig. 3c). However, most genes involved in essential liver function responded at a level correlated to the extent of injury (Fig. 3b–e, Supplementary Figs. 6–10, Supplementary Data 5). This is consistent with the larger total loss of hepatocytes in the PH model compared to the APAP model (~70% vs. ~10%, respectively), resulting in a greater need for functional compensation. These findings highlight not only the rapidity of hepatic

functional adaptation but also the plasticity of hepatocytes across the liver lobule.

**Proliferation is inversely correlated with functional adaptation.** Liver regeneration research has traditionally focused on hepatocyte proliferation[4–7]. It is unknown whether actively dividing hepatocytes can equally contribute to functional compensation. We observed a down-regulation of many hepatic function genes during activation of the proliferative response (A24, P48, Fig. 3b–e). Therefore, we identified cells that became transcriptionally active for cell cycle genes in the scRNA-Seq dataset (Fig. 4a), and analyzed hepatocyte-specific transcript output compared to those cells at all time points that are not cycling. Compared to non-cycling cells (NC), there was a significant down-regulation of the Hepatocyte Signature Score in cycling cells (CC) in both injury conditions (Fig. 4b, c, Supplementary Fig. 11, Methods). DEG revealed substantial differences between cycling and non-cycling cells. CCs expressed many classic cell proliferation markers and exhibited down-regulation of many hepatic function genes (Fig. 4d, Supplementary Data 7). Proliferating hepatocytes in general scored lower for hepatocyte markers. This was corroborated in individual cells by co-staining for proliferating cell nuclear antigen (PCNA) and either the glucose transporter *Slc2a2* or serum protein *Alb* by smFISH (PCNA IF/smFISH, Fig. 4e). Additionally, the cell cycle inhibitor p21 is upregulated following acute injury leading to the observed delay in cell proliferation[27]. scRNA-seq data corroborated this previously described observation (Fig. 4f). Further, smFISH analysis of both p21 and functional compensatory genes (Fig. 4g, Supplementary Fig. 11) show a correlation between p21 expression and the functional compensatory response. Taken together, these data suggest that the functional compensatory response is strongest in non-proliferating hepatocytes that are actively inhibited from entering the cell cycle.

To identify pathways and potential upstream regulators involved in cell cycle activation, we performed GSA over DEG calculated between CCs and NCs from A24 and P48, revealing an upregulation of cell cycle-related pathways. Further, there was a strong enrichment for Wnt/β-catenin-related pathways (Supplementary Fig. 11). It has been shown extensively that Wnt signaling is involved with normal hepatocyte turnover and liver regeneration[10,12–24]. These Wnt factors are derived from the endothelium and contribute to the activation of hepatic progenitor cell genes (*Axin2*, *Tbx3*, and *Sox9*)[17,28]. We observed upregulation of *Axin2*, *Tbx3*, and *Sox9* in each acute injury model, with expression reaching multiple cell layers into the midzone (Supplementary Fig. 12). These data support activation of Wnt/β-catenin signaling in response to injury, which may be co-localized with the induction of genes associated with hepatic plasticity[17,29].

**Wnt mediates functional compensation in addition to proliferation.** Given the demonstrated role of Wnt/β-catenin signaling in establishing liver zonation and proliferation during liver

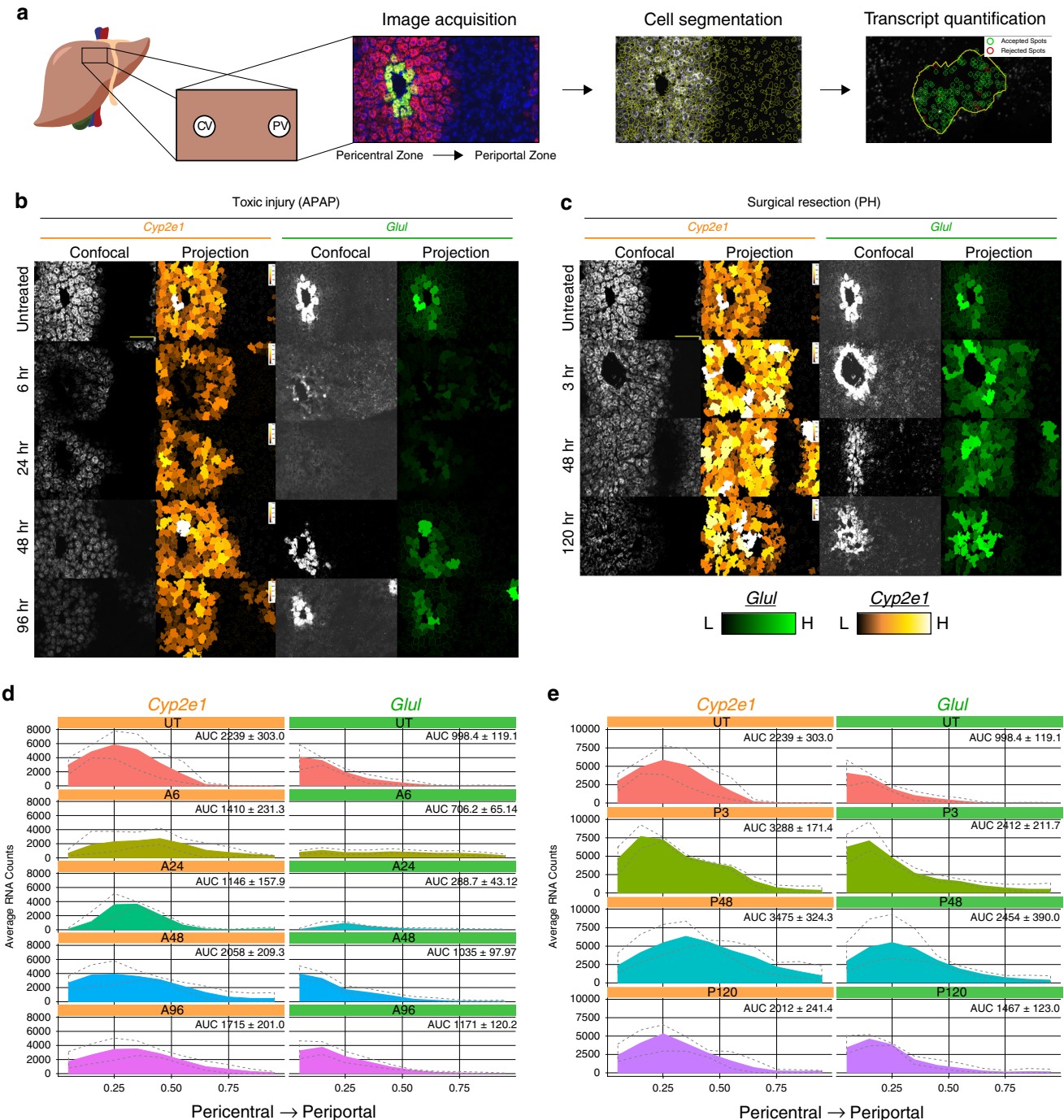

**Fig. 2 Functional compensation of hepatocytes following acute liver injury. a** Schematic for staining and image quantification. **b**, **c** Images of liver section showing pericentral markers *Cyp2e1* and *Glul* for untreated and each APAP-treated (**b**) or PH-treated (**c**) time point (left column). Cell outlined and colored by number of *Cyp2e1* transcripts (brown, low; white, high) for each condition (middle column). Cell outlined and colored by number of *Glul* transcripts (black, low; bright green, high) for each condition (right column). Scale bars are 100 μm. **d**, **e** Quantification of gene expression intensity across the lobule for *Cyp2e1* and *Glul*. 90% of area under the curve (AUC) for UT is to the left dashed red line. treated (**d**) and PH-treated (**e**). Source data provided as a Source Data file.

regeneration, we investigated whether Wnt signaling might activate adaptation of already present hepatocytes to maintain essential hepatic function[18,30–36]. scRNA-seq data corroborated previous observations of increased Wnt activity in proliferating hepatocytes (Fig. 5a)[22,33,37]. Further, it revealed activation of Wnt target gene expression in the majority of hepatocytes in both the APAP and PH models, preceding the onset of cell proliferative activity (A6 and P3, Fig. 5b).

To identify if the hepatic compensatory response following injury is dependent on Wnt/β-catenin signaling, we evaluated conditional Wntless (Wls) KO mice with impaired Wnt processing and secretion as well as hepatocyte-specific β-catenin KO mice after acute injury (Fig. 5c)[18,38] (Supplementary Fig. 13). This strategy enabled identification of the cellular source of secreted Wnt ligands responsible for the functional compensatory response. Loss of either endothelium- or macrophage-derived Wnt secretion

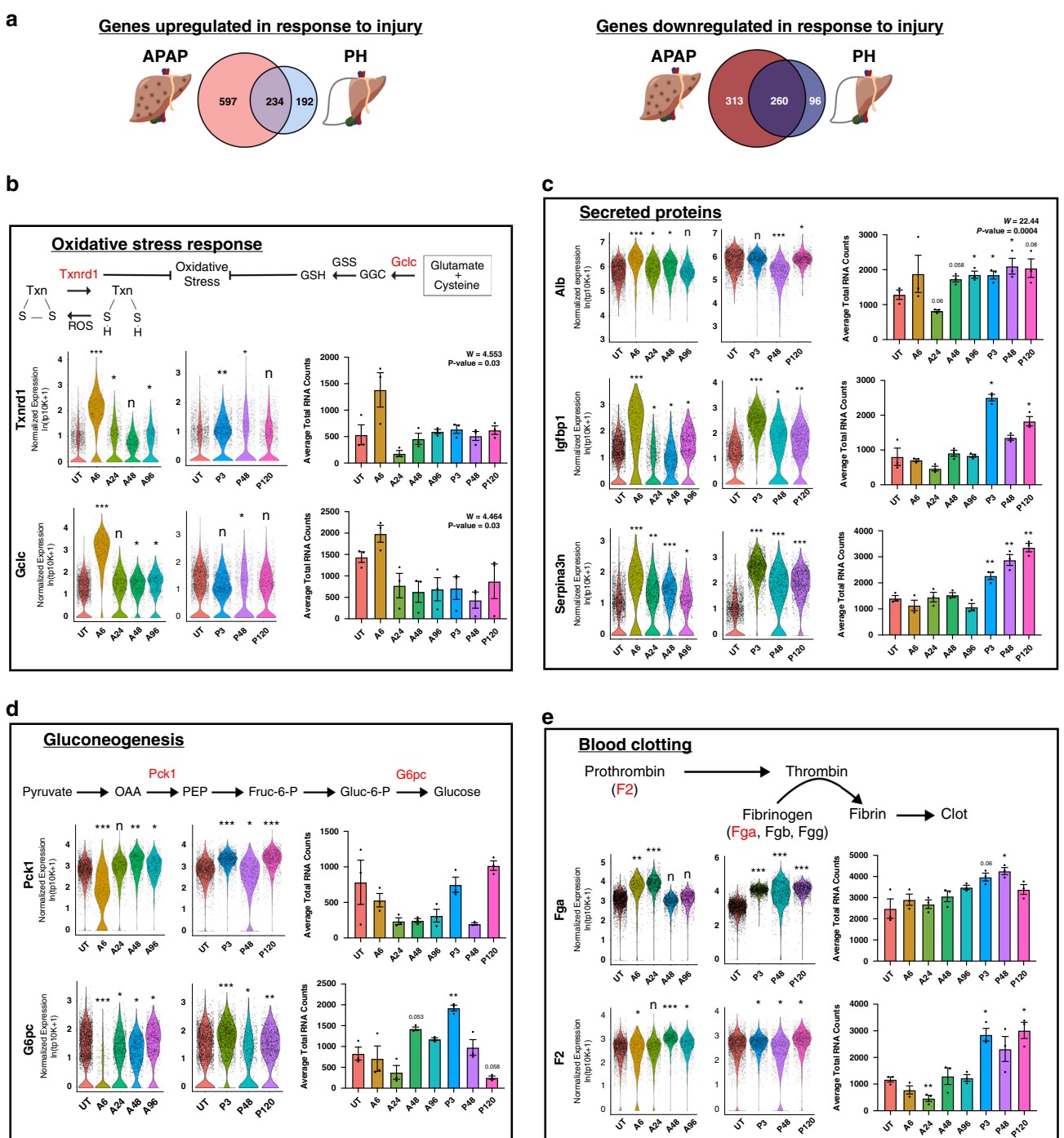

**Fig. 3 Shared and unique gene expression responses in acute liver injury models. a** Venn diagram showing genes significantly changed in response to APAP and/or PH treatment compared to untreated. **b** Expression of oxidative stress response genes (*Txnrd1* and *Gclc*) significantly upregulated in APAP treatment response. **c–e** Expression of genes representing specific hepatic functions (**c**, secreted proteins; **d**, gluconeogenesis; **e**, blood clotting). scRNA-seq data (left) presented as violin plots. smFISH quantification (right) shown as bar plot with individual replicates presented as dots ($n = 3$). Error bars are s.e.m., $P < 0.05$ (*), $< 0.005$ (**), $< 0.0005$ (***), and $< 0.0001$ (****) calculated using unpaired $t$ tests with Welch's correction (two-tailed) comparing untreated group (UT) to all other experimental groups. Source data provided as a Source Data file.

resulted in a down-regulation of hepatic function genes at baseline consistent with the role of Wnt signaling in hepatic function[33,39,40] (Fig. 5d, e). Importantly, EC-*Wls*-KO mice can still transcriptionally compensate following PH but have altered cell proliferation (Fig. 5d, e and Supplementary Fig. 14c)[18]. In contrast, loss of Wnt processing in macrophages of Mac-*Wls*-KO mice resulted in down-regulation of all examined hepatocyte genes after injury with

complete loss of transcriptional compensation following PH. Proliferative capacity was modestly decreased, yet overall maintained (Fig. 5d, e and Supplementary Fig. 14)[41]. Resident hepatic macrophage numbers and geographic distribution were similar between groups (Supplementary Fig. 14b). In addition, combined IF and smFISH staining revealed increased *Glul* transcript level within ~30 μm (average diameter of a single hepatocyte) of F4/80+

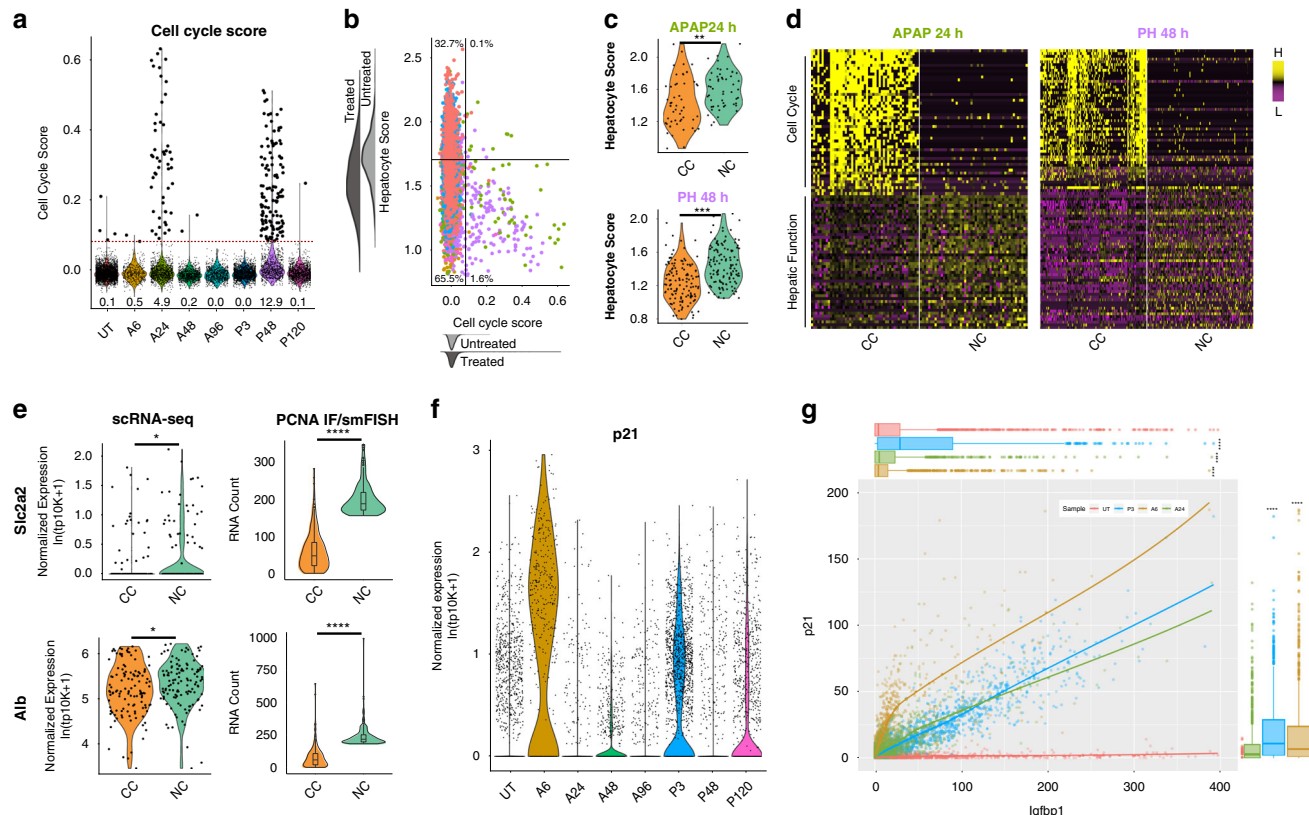

**Fig. 4 Identification and characterization of proliferating hepatocytes. a** Violin plot of cell cycle score across all samples. Cycling cells (CC, larger black dots) are identified as having a cell cycle score two standard deviations above average (dashed red line). Percentage of cycling cells in each condition listed below each violin. **b** Scatter plot of Hepatocyte Score versus Cell Cycle Score. Horizonal line represents average Hepatocyte Score calculated over all untreated cells. Vertical line represents two standard deviations above the average cell cycle score. **c** Violin plots on Hepatocyte Score for all APAP 24 h cycling cells (CC) and an equal number of non-cycling cells (NC) from APAP24 (top) and the same for PH48 CC and NC (bottom). scRNA-seq effect size by absolute value of Cohen's $d > 0.2$; *, $d > 0.5$; **, $d > 0.8$; ***. **d** Heatmap of marker genes of CC and NC in APAP 24 h and PH 48 h. **e** Violin plots of *Alb* and *Slc2a2* in CC and NC cells as measured by scRNA-seq (left) and by co-staining of PCNA and smFISH within tissue sections from P24 (right). scRNA-seq effect size by absolute value of Cohen's $d > 0.2$; *, $d > 0.5$; **, $d > 0.8$; ***. smFISH data tested using an unpaired *t* test with Welch's correction (two-tailed; ****, $P < 0.0001$). **f** Violin plot of p21 expression in scRNA-seq dataset. **g** Quantification of RNA counts of p21 versus Igfbp1. Box plots represent data for both axes. Source data provided as a Source Data file.

macrophages (Supplementary Fig. 14d, e). The number of detected transcripts decreased substantially with increased distance from macrophages (Supplementary Fig. 14f). These data reveal a dichotomy of the function of Wnt signaling within the regenerating liver, where macrophage-derived Wnts are required for the observed functional compensation following PH, while endothelium-derived Wnts are more important for the cell proliferative response. In contrast, livers of Mac-Wls-KO mice maintained the ability to functionally compensate following APAP injury (Fig. 5d, e), suggesting an alternate injury-dependent mechanism for the functional compensatory response.

To investigate the role of intact β-catenin activity on functional compensation, hepatocyte-specific β-catenin KO mice were subjected to acute liver injury (PH) (Supplementary Fig. 13), which revealed that functional compensation is β-catenin-dependent for select genes, such as *Alb*, *Glul*, and *Cyp2e1*. However, *Arg1* is still compensatorily expressed even following β-catenin loss. These findings indicate that both canonical and non-canonical Wnt signaling may play a role in functional compensation of select genes.

## Discussion

The liver uniquely maintains complex metabolic function throughout injury and subsequent regeneration to enable survival of an organism[42,43]. It has long been thought that the liver has sufficient functional reserve to maintain these functions through excess baseline capacity[43–47], but the exact hepatic reserve capacity has been mostly a theoretical concept. Liver injury induces a regenerative response where functionally active hepatocytes are the major contributor to cellular regeneration. Turnover of hepatocytes in the uninjured organ is typically slow, with the entire liver being repopulated by new hepatocytes after ~ 1 year[17,48]. The liver can quickly respond to an acute insult, however, through activation of a regenerative response. Liver regeneration within the mouse model shows a peak of hepatocyte proliferation between 30 and 36 h for both PH and APAP-induced injury[49,50]. Cell cycle genes are activated well before hepatocyte proliferation begins (priming phase) following injury[42,43,51]. Cell cycle inhibitors, such as p21 and p27, however, are concurrently upregulated early in liver regeneration and block progression of hepatocytes into the cell cycle[27,52]. It has been speculated that this co-expression of both stimulators and repressors of the cell cycle aides in the control of liver regeneration to a precise endpoint[42].

Here, we discover a mechanism by which the liver maintains essential function through transcriptional compensation prior to the proliferative response. Hepatocytes upregulate transcription of important liver genes, typically by adapting gene expression patterns extending beyond homeostatic zonal boundaries.

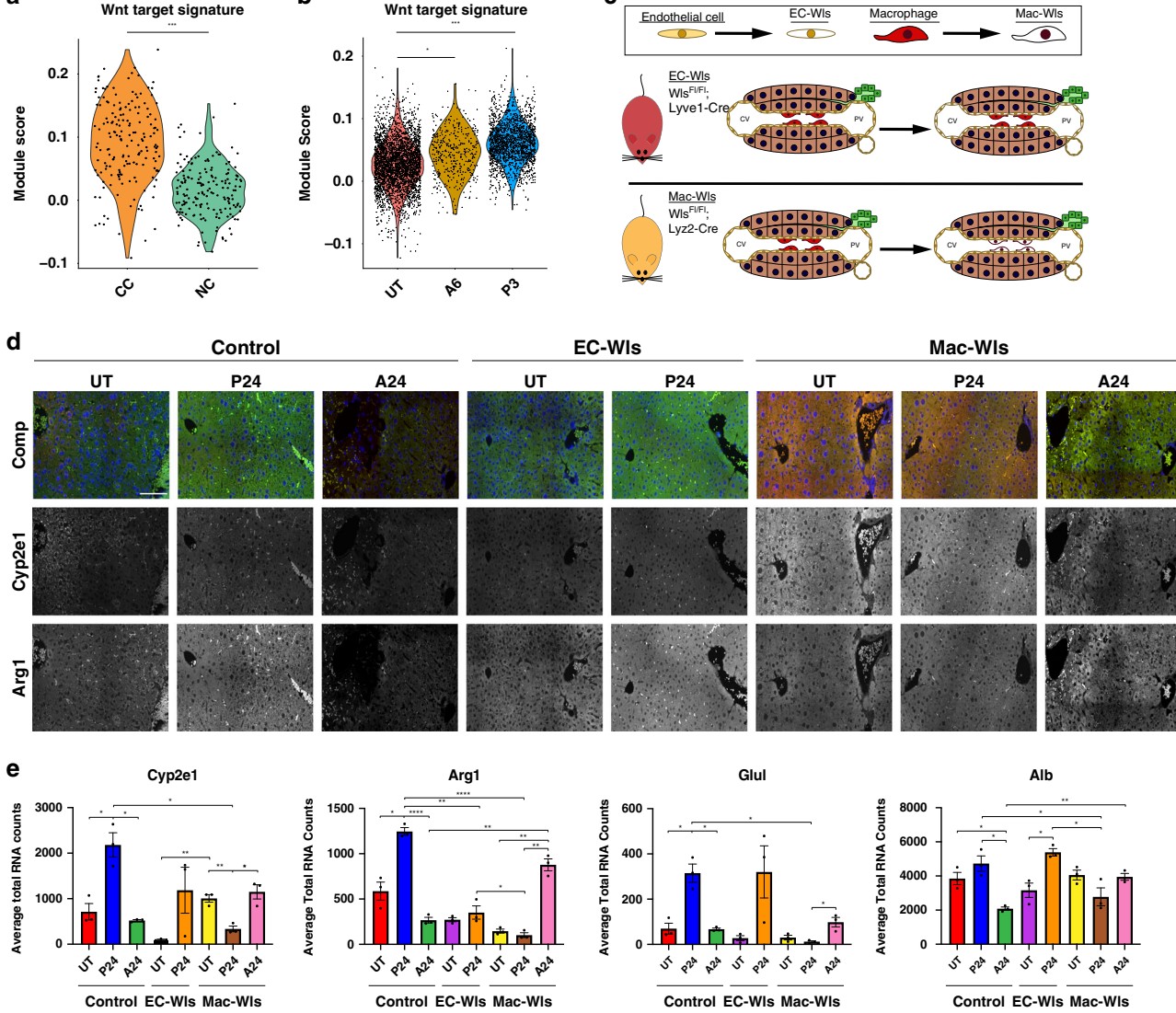

**Fig. 5 Contribution of Wnt signaling to functional compensation of hepatocytes.** Wnt target gene expression score over cycling cells (CC) and non-cycling cells (NC) from A24 and PH48 (**a**) and hepatocytes grouped by treatment condition (UT, A6, and P3) (**b**). **c** Wnt knockout mouse models. **d** Hepatocyte marker expression (*Alb* and *Arg1*) in untreated, P24, and A24 for wild type (Control), endothelial cell Wntless KO (EC-Wls), and macrophage Wntless KO (Mac-Wtls) by smFISH. Representative image taken from three independent liver lobules. Scale bar is 100 μm. (**e**) Average RNA expression of hepatocyte markers (*Alb*, *Arg1*, *Cyp2e1*, and *Glul*) in untreated, P24 hr, and A24 for WT, EC-Wls, and Mac-Wtls by smFISH. smFISH quantification shown as bar plot with individual replicates presented as dots ($n = 3$). Error bars represent s.e.m., $P < 0.05$ (*), $< 0.005$ (**), $< 0.0005$ (***), and $< 0.0001$ (****) calculated using unpaired $t$ tests with Welch's correction (two-tailed). Source data provided as a Source Data file.

Importantly, many hepatocyte function genes are expressed or upregulated predominantly in non-proliferating hepatocytes, while those cells that enter cell cycle express hepatocyte function genes at lower levels. This suggests the presence of a division-of-labor between hepatocytes contributing to the maintenance of hepatic function and those responsible for the proliferation response and a return of cellular mass. Further investigation is needed to determine if select sub-populations of hepatocytes exist that contribute to each aspect of the response or if this dichotomy is present due to the differential availability of energy and cellular building blocks. Collectively, our data define a compensatory phase following liver injury, thereby complementing previous studies which have elegantly established the field of liver regeneration by focusing on the hepatocyte proliferative response[4–8].

We define a dual role for Wnt/β-catenin signaling in liver regeneration: it not only promotes cell proliferation and cellular recovery, as previously demonstrated in multiple studies[10,12–24,35,36],

but is also required for functional compensation (Fig. 6). We identify macrophages, but not endothelial cells, as the main source of secreted Wnts that enable transcriptional compensation, particularly following massive loss of hepatic tissue after PH. Further investigation will be necessary to determine whether other signals are involved in this process dependent on the extent and nature of injury, particularly for APAP toxicity and other forms of chemical liver injury. The contribution of macrophage-dependent Wnts to transcriptional compensation complements other studies, which have highlighted the role of endothelial-derived Wnts to maintain hepatic zonation, as well as both endothelial- and macrophage-secreted Wnts to enable cellular proliferation[15,18,41,53,54]. Our results establish that macrophages, which are responsible for broad inflammatory and immunologic functions[55], are also essential for delivering Wnts locally throughout the entirety of the hepatic lobule, both in midzone and periportal areas, because of their ability to migrate and release Wnt ligands throughout the tissue. It has been

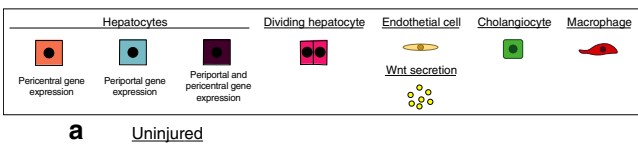

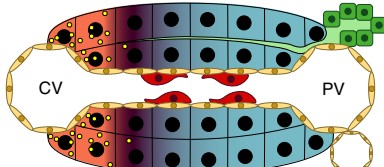

**a** Uninjured

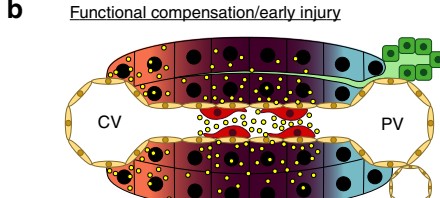

**b** Functional compensation/early injury

**c** Proliferation phase/late injury

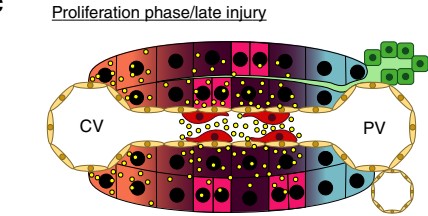

**Fig. 6 Model of hepatocyte response to acute liver injury. a** Wnt secretion from the pericentral endothelium functions in the maintenance of the pericentral gene expression gradient in normal, quiescent liver. **b** Wnt secretion from macrophages aids in functional compensation of midzonal and periportal hepatocytes during the pre-proliferation phase of acute liver injury. **c** Wnt secreation is essential for both functional compensation and activation of the proliferative response during regeneration. Compensating hepatocytes contribute to a maintenance of hepatic function, whereas proliferating hepatocytes selectively down-regulate a subset of hepatic genes.

previously shown that an increased amount of Wnt ligands spanning many members of the family are detectable across the liver lobule during liver regeneration[18]. In particular, Wnts 2b and 9 are primarily upregulated in endothelial cells and macrophages. Further investigation will be needed to determine whether specific Wnt ligands initiate the functional compensatory response during liver regeneration or if all ligands can contribute equally. Additionally, the roles of both canonical and non-canonical Wnt signaling to control compensation of individual genes and specific cellular functions need to be elucidated. Our findings further highlight the potential of the Wnt/β-catenin pathway as a therapeutic target in acute liver failure and other liver pathologies, where maintenance of liver function is essential. Future studies will be needed to identify specific Wnt ligands to promote liver function, regeneration, and survival in regard to multiple pathologies that result in acute liver failure.

## Methods
**Animals**. Three-month-old, male, C57BL/6J mice, purchased from Jackson Laboratories (Bar Harbor, ME, USA), were used in acute liver injury studies (APAP and PH). β-catenin KO studies were conducted using $Alb$-cre$^{+/-}$;Ctnnb1$^{flox/flox}$ mice. Wntless KO studies were conducted using Lyve1-cre$^{+/-}$;Wls$^{flox/flox}$ (endothelial cell, EC-Wls) and Lyz2-cre$^{+/-}$;Wls$^{flox/flox}$ (macrophages, Mac-Wls). All animals were housed in Association for Assessment and Accreditation of Laboratory Animal Care—accredited facilities under a standard 12-h light/dark cycle at 71 °F and 30–70% humidity with access to chow and water ad libitum.

Partial hepatectomy studies were approved by the University of Kansas Medical Center and University of Pittsburgh Institutional Animal Care and Use Committees (IACUC). APAP studies were approved by the University of Pittsburgh Medical Center and Brigham and Women's Hospital IACUCs. Wnt studies were approved by the University of Pittsburgh Medical Center IACUC. All relevant guidelines for work with mice were adhered to during this study.

**Acetaminophen (APAP) exposure**. Mice were fasted 12 h before administration of APAP. APAP was dissolved in warm 0.9% saline, and mice were injected with 300 mg/kg APAP, i.p. Food was returned to the mice after APAP treatment. Mice were then used for isolation of primary hepatic cells for single-cell RNA-sequencing or tissue harvest for further downstream analysis.

**Partial hepatectomy**. Partial hepatectomy surgeries were performed as previously described[52]. Mice were euthanized at 3 h, 48 h, and 5 days post-partial hepatectomy by cervical dislocation under isoflurane anesthesia and livers were harvested for downstream analysis. Further, mice were used for isolation of primary hepatic cells at 3 h, 48 h, and 5 days post-partial hepatectomy.

**Tissue harvest**. Untreated ($n = 3$ for each sex) and APAP-treated mice ($n = 3$ for 6, 24, 48, and 96 h following APAP exposure and 3, 48, and 120 h following PH) were euthanized by cervical dislocation following carbon dioxide exposure. A portion of liver tissue was fixed in 10% neutral buffered formalin for 48 h and further processed to obtain paraffin blocks and 5 μm thick sections. A portion of liver tissue was frozen in optimal cutting temperature (OCT) medium and used to obtain 10 μm fresh frozen sections.

**Isolation of primary hepatocytes and non-parenchymal cells**. Mouse hepatic cells were isolated by a modification of the two-step collagenase perfusion method[53]. Cells were isolated from untreated ($n = 3$ for each sex), APAP-treated mice ($n = 2$ for each sex at 6, 24, 48, and 96 h following APAP exposure), and mice subjected to partial hepatectomy ($n = 3$; 3 h, 48 h, and 5 days). The digestion step was performed using Liver Digest Medium (Cat. #17703034; ThermoFisher Scientific; Pittsburgh, PA, USA). Cell suspensions were used immediately for Seq-Well.

**Library preparation and sequencing**. Sequencing libraries were prepared from the single-cell suspension using the Seq-Well method as described in Gierahn et al.[9]. Briefly, a microwell array was loaded with barcoded polyT mRNA capture beads (Chemgenes). Then 200 μl of media containing 15,000 single cells were loaded onto the array and allowed to settle into the wells by gravity. Membrane sealing, lysis, hybridization, reverse transcription, exonuclease digestion, second-strand synthesis, PCR, and library construction by Nextera were all performed as previously described[54]. Resulting libraries were quantified by Qubit and tape station (Agilent), and sequenced on an Illumina NextSeq 500 (UT and APAP samples, 2 arrays per run) or a NovaSeq (PH samples, 10 arrays per run) 30 cycle, paired end sequence reads, single 8 cycle index for NextSeq or dual 8 cycle indexes for NovaSeq.

**Single-cell sequencing data processing**. Sequencing data were demultiplexed and aligned to mm10 with STAR aligner. Libraries were sequenced to an average depth of >48,000 reads per cell per sample. See Supplementary Data 1 for additional sequencing and data quality metrics.

Barcodes with fewer than 400 genes were discarded from the genes x cells data matrix as non-cells, with 16,019 cells remaining. Data were log normalized and TPM-like (base 10,000) normalized and analyzed using the Seurat package in R[54]. The resulting data displayed fairly even nGene and nUMI distributions across each sample type (Supplementary Fig. 2h). We performed a principal component analysis (PCA) and selected the top 13 principal components (PCs) for tSNE dimensional reduction. We then performed shared nearest neighbors (SNN) clustering, and identified 14 distinct clusters in the data (Supplementary Fig. 1c, d). We calculated differential expression across the clusters using Wilcox test in the FindAllMarkers function in the Seurat R package and quantified expression of marker genes for known liver cell populations (Supplementary Fig. 1d, e). We identify nine high-quality hepatocyte clusters, separated by treatment condition; one low-quality hepatocyte cluster with a high percent mitochondrial content and low nGene and nUMI; a kupffer cell cluster; a liver endothelial cell (LEC) cluster; a neutrophil cluster; and a mixed immune cluster, which appears to contain T cells, B cells and monocytes. We calculated a hepatocyte signature score using AddModuleScore in Seurat over multiple highly expressed hepatocyte genes which span the lobule: Apoa1, Glul, Acly, Asl, Cyp2e1, Cyp2f2, Ass1, Alb, Mup3, Pck1, G6pc, Fabp1.

In order to focus on hepatocyte responses, we subsetted our data to include the nine high-quality hepatocyte clusters. Following subsetting, we observed a remaining few cells scoring low on the hepatocyte signature. We filtered out any cells with a Hepatocyte Signature score less than 3 standard deviations below the average as non-hepatocytes (Supplementary Fig. 1e). These non-hepatocyte cells originated primarily from the A6 sample, which had the largest immune infiltration in response to injury and the highest fraction on non-parenchymal cells

in the total sample. The filtered non-hepatocytes are likely non-parenchymal cells incorrectly assigned to a hepatocyte cluster by SNN. Following these filtering steps, we retained 10,833 high-quality hepatocytes for analysis.

**Single-cell sequencing data analysis (hepatocyte data).** We performed dimensional reduction and clustering again on our filtered hepatocyte only dataset. Principal component 1 (PC1) describes 46.9% of and captures technical variation (nGene, nUMI) in the data (Supplementary Fig. 2e). This is not surprising for a dataset comprised of a single-cell type. Each of our treatment conditions scores similarly on PC1 (Supplementary Fig. 2b). PC2 partly captures pericentral–periportal variation. We identify zonally restricted genes in PC2 loadings (Cyp2e1, Cyp1a2, Gstm3; Cyp2f2). We also note periportal–pericentral variation captured in PC4.

To more clearly visualize pericentral–periportal variation, we scored cells on this pericentral–periportal metric. To generate a list of pericentral genes, we calculated gene-by-gene correlations with Cyp2e1, a canonical pericentral gene. To generate a periportal gene list, we selected genes positively correlated with Cyp2e1, and to generate a periportal gene list, we selected genes negatively correlated with Cyp2e1 (Supplementary Data 8). To generate a list of genes to be used for our signatures, we considered all genes with a Cyp2e1 correlation >0.3 for PCHSig and a Cyp2e1 correlation < −0.3 for PPHSig. From the genes falling within this range of values, we selected moderately expressed genes with large variability in expression across the dataset, removing lowly expressed genes and genes expressed in small numbers of cells. Positive correlations with Cyp2e1 range from 0.823 (Cyp2c29) to 0.356 (Ang); negative correlations with Cyp2e1 range from −0.569 (Cyp2f2) to −0.311 (Serpina12). The 0.3/−0.3 cutoff is more than 3 standard deviations above/below the mean of all gene correlations with Cyp2e1 (mean = −0.01280654; mean + 3sd = 0.2605311; mean − 3sd = −0.2861442). We then calculated the pericentral hepatocyte (PCH) score and periportal hepatocyte (PPH) score using AddModuleScore for these genes. We then confirmed that PCH Score and PPH Score are inversely correlated as expected. We observe a pericentral–periportal gradient across PC2 using these scores (Supplementary Fig. 2). To generate a single score that captures pericentral–periportal character, we subtracted the PCH Score from the PPH Score to create the PPH-PCH Score, in which pericentral hepatocytes will score negatively and periportal hepatocytes will score positively.

We also generated PCH and PPH scores from the zonated hepatocyte genes data presented in Halpern et al., Table S3. Briefly, we selected non-randomly zonated genes with a q-value < 0.01, removed lowly expressed genes with a total average expression lower than the average over the dataset, and split the genes into percentral or periportal markers based on relative zonal expression. This resulted in 110 genes for PCH and 96 genes for PPH. We calculated module scores with these gene lists for the dataset. The PCH_Halpern Score did not capture loss of PCH in the A6 sample as well as the PCH Score calculated above with the Cyp2e1 correlation gene list. The PPH_Halpern Score did not distribute the cells as evenly over the distribution as the correlation score. These Halpern scores were generated from a different dataset, which may contribute to their lesser ability to capture relevant signal in our dataset. Despite these differences, the PCH and PPH correlation-derived scores correlate well with the Halpern-derived scores (PCH: R2 = 0.567, p < 10e−5; PPH R2 = 0.399, p < 10e−5, Supplementary Fig. 2i, j).

To better visualize the data, we performed tSNE dimensional reduction. Hepatocytes from all samples look rather similar in lower PCs which describe shared variation, such as technical differences or cross-lobule variation, while the higher PCs capture inter-sample variation. We calculated percent variation captured per PC and generated an elbow plot to determine the correct number of PCs to use in further analysis. We selected the top 13 PCs to include in our analysis, which well separated samples by treatment condition and did not appear to be driven by technical artifacts. We observe a technical gradient across each cluster (which is orthogonal to the pericentral–periportal gradient across each cluster), but the clusters themselves do not appear technically driven (Supplementary Fig. 2h).

Heatmap genes were found using FindAllMarkers in Seurat, Wilcox test, min. percent = 0.10, thresh.use = 0.25. Mitochondrial and hemoglobin genes were removed from the list prior to heatmap plotting.

Shared and unique by injury model gene lists were assembled by combining DE results across all time points for each injury. We ran differential expression using Wilcox test in the FindAllMarkers Seurat function between each treatment condition (A6, A24, A48, A96, PH3, PH48, PH120) individually and untreated (UT) (Supplementary Data 2). We then combined results across all time points within each injury model to obtain a list of all DEGs from any time point in APAP experiments (A6, A24, A48, A96) and PH (PH3, PH48, PH120). A small number of genes were up at one-time point, but down at another. In these cases, the gene was retained in the list (up- or down-regulated) with the largest magnitude average log fold-change to capture to more significant change in expression.

We ran pathway analysis on the composite DE results using the piano R package. Reference gene sets were downloaded from MSigDB (Broad Institute). We used geneSetStat = "fisher", adjMethod = "fdr", and signifMethod = "geneSampling". We then parsed the results to identify shared and unique reference gene sets for each injury. Any reference gene set with a q-value greater than 0.05 was discarded as insignificant. We then identified reference gene sets with significant overlaps with

only APAP and with only PH composite DE results. To focus on truly unique responses, we filtered out any reference gene set from the unique tables which had a q-value < 0.2 for the other injury model. We then identified shared responses by compiling all reference gene sets with a q-value of < 0.05 in both APAP and PH. Selected reference gene set −log(q) are plotted in Fig. 3.

To identify cycling cells in the data, we calculated Cell Cycle Score using AddModuleScore in Seurat over the cell cycle markers[56]. We classified cells with a Cell Cycle Score 2 standard deviations above the average as cycling cells (Fig. 4). To better compare cycling and non-cycling cells (CC and NC, respectively), we subsetted the data to create a dataset containing all 51 CCs from the A24 condition and an equal number of NCs also from A24; similarly, we created a dataset containing all 123 CC from PH48 and an equal number of NCs also from PH48. Pathway analysis was done on a DE result obtained from comparing 174 CC from A24 and PH48 against an equal number of NCs from these time points. Piano was run as described above. Fig. 4 plots −log(q) values for selected reference gene sets with a q-value < 0.05. Wnt Target Labbe Sig was calculated using AddModuleScore and the reference gene set LABBE_WNT3A_TARGETS_UP which was identified as significant in Piano gene set enrichment analysis.

**Immunohistochemical analysis.** Histology was performed by the histology core at Beth Israel Deaconess Medical Center using standard procedures and automated workflow. Samples were processed and embedded following fixation in 10% neutral buffered formalin for 48 h. Samples were embedded in paraffin and sectioned at 5 μm thick. Immunochemistry was performed on a Leica autostainer (Leica Biosystems) with enzyme treatment (1:1000) using standard protocols. The antibody used for assessment of cell proliferation was PCNA (Cell Signaling, Cat. 13110, 1:800), and cell death was ApopTag Peroxidase In Situ Apoptosis Detection Kit (Millipore, Cat. # S7100). Macrophages were stained using the anti-F4/80 (Cell Signaling, Cat. 70076, 1:50). Sections were then counterstained with hematoxylin, dehydrated, and film cover slipped. A minimum of four representative images was captured per slide using the Zen 2 v2.0 software package. TUNEL-positive area, PCNA-positive cells, and F4/80-positive cells were measured and averaged across the four images for each sample using ImageJ v2.0.

**Single-molecule fluorescent in situ hybridization (smFISH).** smFISH was conducted using RNAscope technology (RNAscope Fluorescent Multiplex Kit; Cat. # 320850; Advanced Cell Diagnostics; Neward, CA, USA). Fresh frozen sections (10 μm thick) were used following the manufacturer's guidelines. Probe sets were designed by the manufacturer and can be found at acdbio.com/catalog-probes. Probe sets were chosen for genes that span multiple classic hepatic functions, such as secreted proteins, gluconeogenesis, blood clotting, ion homeostasis, and metabolism. and that were among the top 100 DEGs in the analysis of the scRNA-seq data. A 6 × 6 40x field was captured of a 10 μM z-stack (0.5 uM per slice) using the NIS-Elements v5.11 software package. This resulted in multiple liver lobules available for analysis within a single section. Images were cropped to the size of a single liver lobule and cellular outlines were defined using CellProfiler v2.2[55]. smFISH signal was then quantified using FISH-quant v3[57]. Post-processing of mRNA detection was performed with custom-written Python scripts (available at https://bitbucket.org/muellerflorian/pyfishquant/)[57]. Pseudo-color images of transcript abundance were generated by setting the pixel values of each segmented cells to its corresponding transcript level. To determine the spatial expression gradients relatively to the central vein, we manually outlined the vein as a polygon. This polygon served as a reference point to count RNAs in concentric rings. These counts were lastly renormalized by the ring area contained within the image. AUC (area-under-curve) for the RNA counts across the individual lobules was calculated using GraphPad Prism (v8.3.0) with the average AUC presented with calculated standard deviation. The values are presented as "Average Total RNA Counts".

**Statistical analysis.** We calculated P values for shifts in gene expression or module scores using the Wilcox test, Bonferroni corrected for multiple testing. Gene set enrichment results in piano were calculated using Fisher's test and the gene sampling method and corrected by FDR. P values for average RNA expression (smFISH) and IHC counts (PCNA, F4/80, and TUNEL) were calculated using unpaired, two-tailed, t tests with Welch's correction. Histograms were tested using the Kolmogorov–Smirnov test. Specific tests used for each figure are listed in the figure legends with represented P values. Statistical analysis of IHC and smFISH quantification was performed using Graphpad Prism 8 v8.2.1.

## Data availability
Single-cell RNA-seq data are available through the NCBI's Gene Expression Omnibus (GEO Accession GSE136679) Data are available upon reasonable request. Source data are provided with this paper.

## Code availability
Custom software for analysis of smFISH data along with descriptions of functionality and sample data can be found at https://github.com/muellerflorian/walesky-rna-loc-liver.

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

## Acknowledgements

C.W. is supported by F32DK111151, the Cholangiocarcinoma Foundation Research Fellowship, and the American Liver Foundation Charles Trey, MD Memorial Post-doctoral Research Fellowship. W.G. is supported by R01DK090311, R01DK105198, R24OD017870, and the Claudia Adams Barr Program for Excellence in Cancer Research. W.G. is a Pew Scholar in Biomedical Sciences. A.K.S. was supported, in part, by the Searle Scholars Program, the Beckman Young Investigator Program, a Sloan Fellowship in Chemistry, the NIH (1DP2GM1194192 & RM1HG0061931), and the MIT Stem Cell

Initiative through Fondation MIT. U.A. is supported by R01DK098414. S.P.M is supported by NIH/NIDDK P30DK120531, Pittsburgh Liver Research Centre Grant, and NIH/NIDDK (DK62277, DK100287, and CA204586). S.P.M. is the Endowed Chair for Experimental Pathology, University of Pittsburgh School of Medicine.

## Author contributions

C.M.W., K.E.K., A.K.S., and W.G. conceived the project and designed the experiments. C.M.W. and C.L.W. carried out all wild-type APAP studies. U.A., C.M.W., and J.H. carried out wild-type partial hepatectomy studies. C.M.W., K.E.K., C.L.W., J.H., B.K., and I.F. carried out all scRNA-seq studies. K.E.K. performed all scRNA-seq data processing. K.E.K. and C.M.W. performed scRNA-seq data analysis. C.M.W. and C.L.W. performed all IHC and smFISH experiments and analysis. F.M. performed software development for smFISH analysis. S.K. and S.P.M. performed all Wnt studies including APAP and PH for these models. C.M.W., K.E.K., A.K.S., and W.G. wrote the manuscript with the aid of all co-authors.

## Competing interests

W.G. is a consultant for Camp4 Therapeutics. A.K.S. reports compensation for consulting and/or SAB membership from Merck, Honeycomb Biotechnologies, Cellarity, Cogen Therapeutics, Ochre Bio, and Dahlia Biosciences. S.P.M has an advisory board role for Surrozen. C.M.W., K.E.K., C.L.W., J.H., B.K., I.F., F.M., S.K., and U.A. have no competing interests to declare.
