## [Peer Review File · Nature Communications]

Reviewers' comments:

Reviewer #1 (Remarks to the Author):

In this manuscript, Walesky et al address the question of how the liver maintains its function during massive injury. This is an important question that has been less studied compared to the mechanism of liver regeneration. They address this question using a combination of single cell RNA sequencing and single molecule fluorescence in situ hybridization using two different models of liver injury. They show that there is transcriptional adaptation by hepatocytes after injury, particularly in the subset of hepatocytes that are not actively cell cycling. Importantly, they identify a novel role of macrophages and macrophage derived Wnt signals in maintaining this adaptive process.

Overall, I found this to be a very well designed study that leverages the development of new single cell transcriptomic and single molecule FISH imaging techniques to dissect out the heterogeneous response of hepatocytes after injury. The identification of macrophage derived Wnts as important to this process is a significant finding. I have a few comments that I'd like the authors to address.

1. One of the main strengths of single cell transcriptomic studies is the ability to identify transcriptomically distinct subpopulations which are masked in bulk RNAseq analysis. In this manuscript, the authors primarily used single cell RNAseq data as bulk RNAseq data, except for distinguishing cell cycling and non-cycling hepatocytes. Can you identify transcriptionally distinct clusters/subpopulations of hepatocytes within each post-injury time point? Presumably the inductive signals such as macrophage-derived Wnts induce hepatocytes along the lobule to express a new adaptive gene expression profile after injury. For example, do the mid-zonal cells that show de novo expression of pericentral genes after APAP injury have a distinct gene expression profile? I recognize that the single cell RNAseq data is limited by read depth and number of genes captured so this may be difficult to discern.
2. Do the non-cycling cells that show increased transcription of functional hepatocyte genes show lower levels of expression of cell cycle inhibitors such as p21 and p27?
3. Can you comment on which Wnts are produced by macrophages after injury? Are these cells spatially localized to a particular region of the liver lobule? Since Wnt proteins are hydrophobic and do not diffuse far, they typically act in a paracrine fashion. Can you show the spatial relationship between Wnt-producing macrophages and responding hepatocytes? Particularly the non-cycling hepatocytes?

Reviewer #2 (Remarks to the Author):

In this manuscript by Walesky et al., the authors employed single-cell RNA-seq and smFISH in two acute liver injury model and identified: (1) functional compensation precedes cellular proliferation; (2) decoupling of hepatocytes proliferation and function; and (3) WNT signaling from macrophages plays an important role in hepatocyte adaptive reprogramming. Conceptually, this is an important application of novel single-cell methodologies to study well established models of liver injury-regeneration. In practice, however, the authors did not take full advantage of the extensive datasets to perform more refined analyses. Below are my specific comments:

(1) In Fig. 1f, it seems that Glul transcript level and the percentage of Glul positive cells are not significantly changed in all time points of APAP. However, in smFISH experiment shown in Figs. 2b and 2d, the percentage of Glul positive cells is increased. Is the discrepancy due to the difference in sensitivity between single-cell RNA-seq and smFISH? Or during single-cell RNA-seq sample preparation, there is a bias in cell representation? In a bigger picture, what are the percentage of pericentral and periportal cells captured in the single-cell RNA-seq experiment? The discrepancies between single-cell RNA-seq and smFISH quantifications are obvious in multiple other instances, e.g. in Figs. 3d and 3e. Can the authors comment of this?

(2) To score the single hepatocytes with pericentral (periportal) signatures based on a dozen of gens is too crude. To further refine the spatial annotation, the authors should consider using the liver zonation markers reported by Halpen et al (Nature. 2017 Feb 16; 542(7641): 352–356).

(3) For gene differential expression analysis, the authors should consider performing time-point specific systematic analyses to reveal gene expression kinetics in addition to grouping all the time points in each treatment together.

(4) The six transcripts the authors chose to display in details in Figs. 3d and 3e seem cherry-picking. What are the selection criteria for choosing these transcripts?

(5) Are the cycling cells in different treatments more biased towards pericentral or periportal?

(6) The graphs in Fig. 4g are hard to interpret. As an alternative way to quantify the relationship between function and proliferation, the authors may consider binarizing the cells as CC or NC based on PCNA staining threshold and then quantify the smFISH labeling of markers in cells of the two different categories, similar as shown in Fig. 4e.

(7) In the experiment performed in Fig. 5c, what is the changes in hepatocyte proliferation following Wis-KO?

Minor points:

(1) The descriptions of Figs. 2d and 2e in the legend are not clear.

- (2) Line 150, the sentence is not complete.
- (3) The colors are switched in Fig. 3C, middle panel.
- (3) Supplementary Fig. 2, no description for panel g.
- (4) Supplementary Figs. 3 and 4 are switched.

Reviewer #3 (Remarks to the Author):

The manuscript presented by C Walesky et al, studies the hepatocyte reprogramming during liver regeneration after two very distinct injuries, APAP injury and Partial Hepatectomy (PH). The authors used single-cell RNA-seq combined with smFISH and mouse models for endothelial- and macrophage-specific Wls deletion (EC-Wls, Mac-Wls, respectively). They described a transcriptional compensation in mostly non proliferating hepatocytes that modified the zonal expression, likely required to maintain the essential liver functions during the repair mechanism. They also proposed that macrophage-derived Wnt signals are key actors of this transcriptional reprogramming. The technical approach of single-cell RNA-seq is well appropriate.

However, even, if I am very inclined to believe that this mechanism should be crucial for liver repair, I am very concerned by the poor quality of the interpretation of some results, the poor quality of the PCNA staining to analyse hepatocyte proliferation (Figure 1b) and by several errors in both the numbering of the figures in the text, and more seriously, in the reversal between the WT and KO in some figures, that do not help for the clarity of the message. The demonstration that macrophage-derived Wnt signals are key actors of this transcriptional reprogramming merits also to be reinforced. Finally, the authors claimed that their work describes a new mechanism, but it has already been described partly by the authors (Preziosi et al, *HepatoL Communications* 2018).

Major concerns.

1. During APAP injury, the authors claimed that the pericentral signature was returned at 24h and take Cyp2e1 and Glul as examples. But data presented in Suppl Figure 3 by smFISH do not support this conclusion. Even, if Glul is expressed at low level across the entire liver lobule, its total expression is far less than in control animals and the compensatory expression is not obvious. The sentence "No significant change in Glul+ hepatocytes at any time point " is more than surprsing.

2. P8, line 165. "These results highlight that zonal transcriptional compensation is independent of the form of liver injury occurring after both zone-specific injury and also after massive cellular loss". It should have been more appropriate to describe precisely in each model of liver injury the zonal transcriptional compensation from the single-cell RNA-seq rather than wrote this sentence after analysing only three zoned genes by smFISH.

3. Figure 3. It is unfortunate that the single-cell RNA-seq have been so poorly presented. I do not understand the interest of establishing a signature shared between the two models which are very distinct and, moreover all along the injury response that is also very distinct in these two models. The validation by smFISH is also questionable, why validating glucose homeostasis with Pck1? In contrast, the identification of the oxidative stress response in the APAP model at 6h is interesting and would have merited more attention.

4. It is not very obvious for me why the authors chose a so complicate method to establish a perivenous and periportal signature knowing that such gene lists have already been described by numerous works, including a comprehensive study using smFISH (Halpern et al, Nature 2007). It is surprising that Glul is not on the perivenous list (Supplemental Table 8).

5. The finding described in Figure 4 are expected and may be presented mostly as supplemental data.

6. The involvement of the Wnt/ β -catenin pathway is an interesting point. However, this part of the work should be greatly improved.

a. Figure Supplemental figure 8, there are numerous errors between the WT and KO groups, for Arg1, Cyp2e1, Glul and not for Alb. I supposed, if not, the conclusions are wrong.

b. Figure Supplemental Figure 7. It is hard for me to believe that Axin2 have a strong increase of expression at 6h in the APAP model. There are very few hepatocytes at this time. The author should check if their smFISH are confirmed by the single-cell RNA-seq analysis.

c. Figure 5. I do not understand how the authors concluded from Figure 5a, b, that there is an increase in the Wnt target signature that preceded cell proliferation. Furthermore, the authors have already published that the Mac-WIs model has no change in liver zonation (Yang Hepatology 2004). Yet, on Figure 5d, e, Glul and Cyp2e1 are clearly decreased in Mac-WIs compared to controls.

The EC-WIs that mimicked the Ctnnb1-KO model have a decrease expression of Arg1 compared to controls, that is surprising as Arg1 is a negative target of Ctnnb1. More importantly, the authors concluded on the analysis of only four genes (Cyp2e1, Arg1, Alb, Glul) that "macrophage-derived Wnts are required for the observed functional compensation". It is rather weak and not very strong.

Moreover, it would have been interesting to study the APAP model with Mac-WIs model.

Point-by-point response to “Functional Compensation Precedes Recovery of Tissue Mass Following Acute Liver Injury” by Walesky et al.

NB Reviewer comments are provided in bold; *responses in blue italics.*

Reviewer #1:

In this manuscript, Walesky et al address the question of how the liver maintains its function during massive injury. This is an important question question that has been less studied compared to the mechanism of liver regeneration. They address this question using a combination of single cell RNA sequencing and single molecule fluorescence in situ hybridization using two different models of liver injury. They show that there is transcriptional adaptation by hepatocytes after injury, particularly in the subset of hepatocytes that are not actively cell cycling. Importantly, they identify a novel role of macrophages and macrophage derived Wnt signals in maintaining this adaptive process.

Overall, I found this to be a very well designed study that leverages the development of new single cell transcriptomic and single molecule FISH imaging techniques to dissect out the heterogeneous response of hepatocytes after injury. The identification of macrophage derived Wnts as important to this process is a significant finding. I have a few comments that I'd like the authors to address.

We thank this Reviewer for the positive reception on the significance of our findings and for the helpful suggestions, which have helped to improve our study.

1. One of the main strengths of single cell transcriptomic studies is the ability to identify transcriptomically distinct subpopulations, which are masked in bulk RNAseq analysis. In this manuscript, the authors primarily used single cell RNAseq data as bulk RNAseq data, except for distinguishing cell cycling and non-cycling hepatocytes. Can you identify transcriptionally distinct clusters/subpopulations of hepatocytes within each post-injury time point?

We do observe diversity among hepatocytes within each post-injury time point, but the cells occupy a continuous distribution of states rather than discrete clusters or subpopulations. Previous reports have also observed a single, continuous group of hepatocytes¹.

Presumably the inductive signals such as macrophage-derived Wnts induce hepatocytes along the lobule to express a new adaptive gene expression profile after injury. For example, do the mid-zonal cells that show de novo expression of pericentral genes after APAP injury have a distinct gene expression profile? I recognize that the single cell RNAseq data is limited by read depth and number of genes captured so this may be difficult to discern.

The zonal location of an individual hepatocyte is difficult to identify from the scRNA-Seq data during injury. During homeostasis, the zonal location can be inferred based on expression on zonally distinct genes¹. Under injury conditions, however, as we have shown, under injury conditions, that the zonal pattern of genes is altered, making it difficult to computationally assign hepatocytes to specific zones. This is one of the major incentives for pursuing single-molecule FISH analysis for confirmation.

2. Do the non-cycling cells that show increased transcription of functional hepatocyte genes show lower levels of expression of cell cycle inhibitors such as p21 and p27?

To comprehensively address this point, we have now include figure panels showing p21 expression in our scRNA-seq dataset (Fig. 4f) as well as biaxial plots of p21 co-expression with compensating genes, such as Igfbp1(Fig. 4g) and Serpina3n (Supplementary Fig. 7g) using smFISH-based co-expression analysis. These data show an increase in p21 expression at early time points (A6 and P3) for both injury models, known to precede cell proliferation. This leads us to conclude that cells undergoing active cell cycle inhibition have the ability to compensate at a higher level.

3. Can you comment on which Wnts are produced by macrophages after injury? Are these cells spatially localized to a particular region of the liver lobule? Since Wnt proteins are hydrophobic and do not diffuse far, they typically act in a paracrine fashion. Can you show the spatial relationship between Wnt-producing macrophages and responding hepatocytes? Particularly the non-cycling hepatocytes?

Our collaborators have published that macrophages, as well as endothelial cells, primarily produce Wnts 2b and 9 during liver regeneration². Other members of the Wnt family, however, such as Wnts 2, 4, 5b, and 16, are also detectable at high levels throughout the liver lobule during recovery. We have now added this information into the main text of the manuscript (pg. 16, line 330).

Reviewer #2:

In this manuscript by Walesky et al., the authors employed single-cell RNA-seq and smFISH in two acute liver injury model and identified: (1) functional compensation precedes cellular proliferation; (2) decoupling of hepatocytes proliferation and function; and (3) WNT signaling from macrophages plays an important role in hepatocyte adaptive reprogramming. Conceptually, this is an important application of novel single-cell methodologies to study well established models of liver injury-regeneration. In practice, however, the authors did not take full advantage of the extensive datasets to perform more refined analyses. Below are my specific comments:

We thank the Reviewer for highlighting the key messages of our study, noting its conceptual advance, and for the suggestions to improve our analysis, which we address below.

(1) In Fig. 1f, it seems that Glul transcript level and the percentage of Glul positive cells are not significantly changed in all time points of APAP. However, in smFISH experiment shown in Figs. 2b and 2d, the percentage of Glul positive cells is increased. Is the discrepancy due to the difference in sensitivity between single-cell RNA-seq and smFISH? Or during single-cell RNA-seq sample preparation, there is a bias in cell representation? In a bigger picture, what are the percentage of pericentral and periportal cells captured in the single-cell RNA-seq experiment? The discrepancies between single-cell RNA-seq and smFISH quantifications are obvious in multiple other instances, e.g. in Figs. 3d and 3e. Can the authors comment of this?

This is an excellent point. We suspect that observed differences between scRNA-seq and smFISH may be due to variation that was introduced by using separate cohorts of mice for the respective experiments or due to differences in the sensitivity of the two used techniques³. scRNA-seq is clearly a powerful tool for characterizing transcriptional changes in complex multicellular systems given its ability to simultaneously sample from the whole transcriptomes of multiple single cells at once. This unparalleled ability provides a broad, unbiased view of transcriptional changes within individual cells. The technology, however, has certain limitations, particularly sampling inefficiencies and depth of sequencing, and requires extensive and careful validation. smFISH, while limited to interrogation of carefully selected genes, provides enhanced sensitivity and spatial information, which are essential for validation and understanding how the cellular phenotypes observed by scRNA-seq are distributed in situ.

Of particular importance here is that genes with a relatively low expression level are more likely be affected by limited sequencing depth in scRNA-seq and thus will not be detected in all expressing cells. Glul is an illustrative example of this as it has relatively little to no expression in hepatocytes outside of a single cell layer surrounding the central vein. During the compensation phase, however, we observe a low level of expression across the entire lobule can be observed by (smFISH) that is not detected in the scRNA-seq data. It is likely that other discrepancies can be explained in a similarly fashion.

scRNA-Seq data provides a broad, overall picture of the transcriptional state of the cell. Given capture and quantitation limitations of scRNA-Seq at the individual gene level, smFISH is the more precise technique to confirm data on an individual gene with spatial resolution. We have added text within the manuscript to highlight this point (pg. 7, line 142).

Additionally, we do not see a bias toward zonal cell populations when examining the scRNA-seq data set. Definitions of pericentral and periportal hepatocytes are relative and arbitrary, but we estimate a roughly even mixture in our scRNA-Seq data. Around 50% of hepatocytes express the pericentral marker Cyp2e1, with some variability by individual

mouse. (*Cyp2e1* expression in A6 is expectedly much less than 50%.) Additionally, examination of our pericentral and periportal signature scores reveals that cells are spread somewhat evenly across these scores.

(2) To score the single hepatocytes with pericentral (periportal) signatures based on a dozen of genes is too crude. To further refine the spatial annotation, the authors should consider using the liver zonation markers reported by Halpen et al (Nature. 2017 Feb 16; 542(7641): 352–356).

*We thank the Reviewer for the suggestion to expand our periportal and pericentral signatures. We generated a new signature from the **Table S3** from Halpern et al.⁵ Here, we removed all genes with a q-value > 0.01, removed lowly expressed genes with a total average expression lower than the average over the dataset, and split the genes into pericentral or periportal markers based on relative zonal expression. This resulted in 110 genes for pericentral (PCH) and 96 genes for periportal hepatocytes (PPH). We calculated module scores with these gene lists for the dataset. Unfortunately, the Halpern-derived PCH signature did not capture loss of PCH in the A6 sample as well as the PCH signature used in the submitted version of the manuscript (Halpern-derived PCH score UT v A6 Cohen's $d=1.3$, Correlation-derived Cohen's $d=1.42$). The Halpern-derived PPH signature did not distribute the cells as evenly over the distribution as our original score (Halpern-derived PPH SD=0.158; Correlation-derived PPH SD=0.515). These Halpern scores were generated from a different dataset, generated using a different technology, which may contribute to reduced utility to capture relevant signal in our dataset. Despite these differences, the PCH scores used in our submission correlates well with the Halpern scores (PCH: $R^2=0.567$, $p < 10e-5$; PPH $R^2=0.399$, $p < 10e-5$). We present this analysis in new Supplementary Figure 2i,j.*

(3) For gene differential expression analysis, the authors should consider performing time-point specific systematic analyses to reveal gene expression kinetics in addition to grouping all the time points in each treatment together.

*This is an excellent point. For clarity of presentation, we present only the combined analysis in the main text. Differential expression analysis results for each individual time point have now been added to the supplement (**Table 3**).*

(4) The six transcripts the authors chose to display in details in Figs. 3d and 3e seem cherry-picking. What are the selection criteria for choosing these transcripts?

We apologize for any confusion and that our selection criteria for different gene functions were not properly articulated in the original version of the manuscript. We have now added text to provide clarity on the selection process within the methods and main text. Briefly, our goal was to include probe sets for genes that span multiple essential hepatic functions and also appeared in the top 100 differentially expressed genes in the scRNA-seq data set for a given injury model. Further, we have now added more smFISH data to our analysis so that all functional categories discussed are represented by multiple genes.

We have also added schematics within the relevant figures to clarify the connection between the selected genes and the functional category. We thank the Reviewer for this excellent suggestion.

(5) Are the cycling cells in different treatments more biased towards pericentral or periportal?

*Cycling cells are more pericentral in the APAP injury model. In partial hepatectomy, cycling cells are balanced between pericentral and periportal patterns (**Supplementary Fig. 7e**).*

(6) The graphs in Fig. 4g are hard to interpret. As an alternative way to quantify the relationship between function and proliferation, the authors may consider binarizing the cells as CC or NC based on PCNA staining threshold and then quantify the smFISH labeling of markers in cells of the two different categories, similar as shown in Fig. 4e.

We appreciate the recommendation on how to improve the clarity of our data presentation. We have now updated this panel to use the suggested binary binning and show the data as violin plots.

(7) In the experiment performed in Fig. 5c, what is the changes in hepatocyte proliferation following Wls-KO?

*We would like to thank the Reviewer for raising this excellent point as our efforts to address it have highlighted the presence of a dichotomy with regards to the function of Wnts derived from the endothelium and macrophages. Our data reveal that the liver in the EC-Wls-KO model loses the ability to proliferate after injury, whereas the Mac-Wls-KO model maintains it to a small degree. Additionally, the EC-Wls KO mice maintain the ability to functionally compensate through increased gene expression while the Mac-Wls does not. This suggests that Wnts derived from the endothelium are largely involved in the proliferation response, while Wnts derived from macrophages predominantly are involved in the functional compensation response. We have now added text to the main manuscript to highlight this important observation (**pg. 13, line 267; Supplementary Fig. 10c**).*

Minor points:

- (1) The descriptions of Figs. 2d and 2e in the legend are not clear.**
- (2) Line 150, the sentence is not complete.**
- (3) The colors are switched in Fig. 3C, middle panel.**
- (4) Supplementary Fig. 2, no description for panel g.**
- (5) Supplementary Figs. 3 and 4 are switched.**

Thank you for highlighting these errors. They have now been corrected.

Reviewer #3

The manuscript presented by C Walesky et al, studies the hepatocyte reprogramming during liver regeneration after two very distinct injuries, APAP injury and Partial Hepatectomy (PH). The authors used single-cell RNA-seq combined with smFISH and mouse models for endothelial- and macrophage-specific WIs deletion (EC-WIs, Mac-WIs, respectively). They described a transcriptional compensation in mostly non proliferating hepatocytes that modified the zonal expression, likely required to maintain the essential liver functions during the repair mechanism. They also proposed that macrophage-derived Wnt signals are key actors of this transcriptional reprogramming. The technical approach of single-cell RNA-seq is well appropriate. However, even, if I am very inclined to believe that this mechanism should be crucial for liver repair, I am very concerned by the poor quality of the interpretation of some results, the poor quality of the PCNA staining to analyse hepatocyte proliferation (Figure 1b) and by several errors in both the numbering of the figures in the text, and more seriously, in the reversal between the WT and KO in some figures, that do not help for the clarity of the message. The demonstration that macrophage-derived Wnt signals are key actors of this transcriptional reprogramming merits also to be reinforced. Finally, the authors claimed that their work describes a new mechanism, but it has already been described partly by the authors (Preziosi et al, *HepatoI Communications* 2018).

*We thank the Reviewer for her/his summary and constructive criticism. The conclusions made within this study help to solidify a novel mechanism for functional compensation, which builds upon the previous literature, such as the manuscript referenced above (Preziosi et al, *HepatoI Communications*, 2018). The referenced study by Preziosi et al primarily focused on the proliferative response following PH in both the EC-WIs and Mac-WIs models. Additionally, it highlighted the spatiotemporal dynamics of the Wnt response among various members of the Wnt family with a minor focus on the expression of select Wnt/ β -catenin targets, such as *Ccnd1* and *Glul*. By comparison, the current study utilizes a global and unbiased approach to identify, describe, and validate a previously undescribed phase of liver regeneration aimed at the maintenance of hepatic function, which precedes the proliferative response that has been the major focus of past literature.*

Below, we address each point in turn.

Major concerns.

1. During APAP injury, the authors claimed that the pericentral signature was returned at 24h and take *Cyp2e1* and *Glul* as examples. But data presented in Suppl Figure 3 by smFISH do not support this conclusion. Even, if *Glul* is expressed at low level across the entire liver lobule, its total expression is far less than in control animals and the compensatory expression is not obvious. The sentence “No significant change in *Glul*+ hepatocytes at any time point “ is more than surprising.

The Reviewer is correct that Glul is expressed at a lower level within the APAP model. Accordingly, we have chosen to carefully present that we observe a “low level” of expression. This conclusion is further supported by the presence of measurable Glul transcripts in a similar number of hepatocytes, as compared to controls, within the scRNA-seq data of the A6 and A24 time points. These cells are present even though necrosis persists in the pericentral region, suggesting that they do not line the central vein and may come from another region of the liver lobule, which we validate with smFISH analysis. Additionally, the increase in Glul expression is more pronounced in the PH model, where the extent of injury is much greater and the need for compensation is increased. This is consistent with many of the other genes that we examined. We apologize for any confusion and hope that this clarification is helpful. We have adjusted the text to highlight this point (pg. 7, line 144).

2. P8, line 165. “These results highlight that zonal transcriptional compensation is independent of the form of liver injury occurring after both zone-specific injury and also after massive cellular loss”. It should have been more appropriate to described precisely in each model of liver injury the zonal transcriptional compensation from the single-cell RNA-seq rather than wrote this sentence after analysing only three zoned genes by smFISH.

Thank you for highlighting this point. We appreciate that this statement may over-interpret the data that has been presented. We have modified the statement within the main text to “hepatocytes have the ability to alter their transcriptional output to maintain the expression of zonally expressed genes otherwise lost due to injury”.

3. Figure 3. It is unfortunate that the single-cell RNA-seq have been so poorly presented. I do not understand the interest of establishing a signature shared between the two models which are very distinct and, moreover all along the injury response that is also very distinct in these two models. The validation by smFISH is also questionable, why validating glucose homeostasis with Pck1? In contrast, the identification of the oxidative stress response in the APAP model at 6h is interesting and would have merited more attention.

*We apologize for the confusion surrounding both our data presentation and gene selection. We have adjusted the figure panels and language for **Figure 3** to highlight similarities in the functional compensatory response when comparing the two models, which appears to center around maintenance of hepatic function (pg. 10, line 193). Further, we now highlight our gene selection process within both the main text and methods. Briefly, we chose Pck1 due to its presence within the scRNA-seq data set as well as its established role in gluconeogenesis³. We have modified the title of the represented hepatic function from “Glucose Homeostasis” to “Gluconeogenesis”, and now include a schematic that illustrates where Pck1 contributes to this function.*

We appreciate and also find the oxidative stress response interesting in the APAP model. However, we did not want to overemphasize this point as it does not necessarily follow

the theme of the rest of the manuscript. We do plan to investigate this further in subsequent studies (pg. 15, line 321).

4. It is not very obvious for me why the authors chose a so complicate method to establish a perivenous and periportal signature knowing that such gene lists have already been described by numerous works, including a comprehensive study using smFISH (Halpern et al, Nature 2007). It is surprising that Glul is not on the perivenous list (Supplemental Table 8).

We apologize for the lack of clarity here. The inability to detect Glul is more than likely a technical limitation of scRNA-seq. Glul is normally expressed in a small number of cells within the normal liver, which decreases the likelihood that one will capture a measurable population when examining the entire liver lobule. Following injury, the expression of Glul is present in an increased number of cells across the liver lobule; however, its expression is relatively low when compared to most other genes, which makes Glul susceptible to dropout and decreases the likelihood that measurable levels of the transcript will be detected. That is why the provided smFISH is important for the interpretation of our datasets.

*We generated a new signature from the **Table S3** from Halpern et al.⁵ Here, we removed all genes with a q-value > 0.01, removed lowly expressed genes with a total average expression lower than the average over the dataset, and split the genes into pericentral or periportal markers based on relative zonal expression. This resulted in 110 genes for pericentral (PCH) and 96 genes for periportal hepatocytes (PPH). We calculated module scores with these gene lists for the dataset. Unfortunately, the Halpern-derived PCH signature did not capture loss of PCH in the A6 sample as well as the PCH signature used in the submitted version of the manuscript (Halpern-derived PCH score UT v A6 Cohen's $d=1.3$, Correlation-derived Cohen's $d =1.42$). The Halpern-derived PPH signature did not distribute the cells as evenly over the distribution as our original score (Halpern-derived PPH $SD=0.158$; Correlation-derived PPH $SD=0.515$). These Halpern scores were generated from a different dataset, generated using a different technology, which may contribute to reduced utility to capture relevant signal in our dataset. Despite these differences, the PCH scores used in our submission correlates well with the Halpern scores (PCH: $R^2=0.567$, $p < 10e-5$; PPH $R^2=0.399$, $p < 10e-5$). We present this analysis in new Supplementary Figure 2i,j.*

5. The finding described in Figure 4 are expected and may be presented mostly as supplemental data.

We agree with the Reviewer that it has long been appreciated that liver function, on an organ-wide level, is maintained during liver regeneration. However, it has not been clear which cells are involved in this response. Importantly, an unanswered question in the field is whether proliferating hepatocytes maintain the same functional capacity as non-proliferating hepatocytes. Here, we have provided evidence that proliferating hepatocytes have decreased transcriptional output with regard to hepatic function genes. Further, we have shown that hepatocytes that are being actively inhibited from entering the cell cycle

(p21+) are the cells that are functionally compensating at the highest level. We believe these conclusions are relevant to warrant inclusion of these data in the main figures of the manuscript. We have added text to highlight these key contributions (pg. 11, line 222; pg. 15, line 306).

6. The involvement of the Wnt/ β -catenin pathway is an interesting point. However, this part of the work should be greatly improved.

a. Figure Supplemental figure 8, there are numerous errors between the WT and KO groups, for Arg1, Cyp2e1, Glul and not for Alb. I supposed, if not, the conclusions are wrong.

We would like to thank the Reviewer for the careful assessment and for pointing out this mistake (for Glul and Cyp2e1) which has now been corrected.

The data for Arg1 has been re-analyzed and is consistent with our initial submission. In short, we see a similar expression pattern to that which was reported for the WIs KO model, where Arg1 expression is no longer inhibited in the cells directly surrounding the central vein. However, the total number of transcripts across the entire lobule has decreased when measured using smFISH quantification.

b. Figure Supplemental Figure 7. It is hard for me to believe that Axin2 have a strong increase of expression at 6h in the APAP model. There are very few hepatocytes at this time. The author should check if their smFISH are confirmed by the single-cell RNA-seq analysis.

To address this concern, we have now repeated the measurement of Axin2 using smFISH. This produced similar results as the scRNA-seq. The APAP model will result in necrosis of the pericentral Axin2+ cells. However, ~80% of the hepatocytes, however, remain in this model with increased expression of Axin2 surrounding the periphery of the injured area. This is consistent with a recently published collaborative study from the Nusse and Wang laboratories where carbon tetrachloride was used to induce pericentral necrosis⁴. The Reviewer provides an excellent recommendation of verifying these results within the scRNA-seq data set. However, Axin2 expression is relatively low and, due to the technical limitations of scRNA-seq, we suspect that it is susceptible to dropout with inadequate detection in the data set.

c. Figure 5. I do not understand how the authors concluded from Figure 5a, b, that there is an increase in the Wnt target signature that preceded cell proliferation. Furthermore, the authors have already published that the Mac-WIs model has no change in liver zonation (Yang Hepatology 2004). Yet, on Figure 5d, e, Glul and Cyp2e1 are clearly decreased in Mac-WIs compared to controls. The EC-WIs that mimicked the Ctnnb1-KO model have a decrease expression of Arg1 compared to controls, that is surprising as Arg1 is a negative target of Ctnnb1. More importantly, the authors concluded on the analysis of only four genes (Cyp2e1, Arg1, Alb, Glul) that “macrophage-derived Wnts are required for the

observed functional compensation”. It is rather weak and not very strong. Moreover, it would have been interesting to study the APAP model with Mac-WIs model.

We appreciate the Reviewer highlighting the inconsistency in the analysis of Glul and Cyp2e1. We have repeated this staining and analysis and find no significant difference between the control and Mac-WIs groups for either of these genes.

Arg1 has previously been described as a negative target of b-catenin, with an expression pattern inverse to that of Glul (i.e., all hepatocytes are positive for Arg1 expression except a single layer of cells surrounding the central vein). EC-WIs KO results in expression of Arg1 in all hepatocytes of the liver, including those surrounding the central vein⁵. We observe consistent results at the transcriptional level by smFISH. Critically, smFISH is a highly quantitative method, and here reveals an overall decrease in the transcript number across the entire liver lobule in the EC-WIs KO model even though all hepatocytes are now able to express the gene. This may be partially explained by a compensatory down-regulation due to the increased number of cells now expressing the gene.

We would like to thank the Reviewer for the valuable suggestion of adding the APAP model to the Wntless studies. This has provided evidence for an alternate injury-dependent mechanism for functional compensation, where macrophage-derived Wnts are important for the compensatory response following PH yet play a lesser role in the response following APAP toxicity. This has opened additional avenues for future investigation into the mechanism of functional compensation in regard to differences in the extent of injury, chemical-induced injury, and injury dependent on oxidative stress.

References

1. Halpern, K. B. *et al.* Single-cell spatial reconstruction reveals global division of labour in the mammalian liver. *Nature* (2017). doi:10.1038/nature21065
2. Preziosi, M., Okabe, H., Poddar, M., Singh, S. & Monga, S. P. Endothelial Wnts regulate β -catenin signaling in murine liver zonation and regeneration: A sequel to the Wnt-Wnt situation. *Hepatol. Commun.* **2**, 845–860 (2018).
3. Shalek, A. K. *et al.* Single-cell transcriptomics reveals bimodality in expression and splicing in immune cells. *Nature* **498**, 236–240 (2013).
4. Zhao, L. *et al.* Tissue Repair in the Mouse Liver Following Acute Carbon Tetrachloride Depends on Injury-Induced Wnt/ β -Catenin Signaling. *Hepatology* **69**, 2623–2635 (2019).
5. Leibing, T. *et al.* Angiocrine Wnt signaling controls liver growth and metabolic maturation in mice. *Hepatology* **68**, 707–722 (2018).

REVIEWER COMMENTS

Reviewer #1 (Remarks to the Author):

I appreciate the additional analysis the authors performed to clarify and bolster the point that non-cycling hepatocytes are more involved in functional compensation during regeneration. The additional findings of the different functions between endothelial-derived Wnts and macrophage-derived Wnts is quite interesting. I still have concerns that the authors were not able to maximize the utility of a single cell-based approach to further delineate heterogeneity within hepatocytes at each of the time points in the study. However, as a whole the authors have shown convincingly novel insights about functional compensation in hepatocytes in response to injury.

Reviewer #2 (Remarks to the Author):

In the revised manuscript, the authors provided more details on their analyses and included additional smRNA-FISH data as well as experimental data on mouse Wntless knockout model. They have satisfactorily addressed all my concerns. My remaining suggestion is to increase the font size for the figures to improve readability.

Reviewer #3 (Remarks to the Author):

I clearly appreciate the improvement of the manuscript that makes the messages clearer.

However, there are still surprising results with the β -catenin-KO model that impact on the conclusions raised by the authors. Furthermore, the discussion should include previous works from other groups and from their own groups on the role of the Wnt/ β -catenin signalling in liver homeostasis and during liver regeneration/injury.

1. First, if I am correct I do not find the description of the method used by the authors to establish the average mRNA expression from the smFISH data. In the data source file, the SD for all the genes are relatively small, although the different values between each three mice of a same group are rather heterogeneous. This is the case for most of the genes studies. I did not put in question the quality of the smFISH data that showed expected results for the different genes studied. Expression

of Glul, Cyp2e1, Arg1, Alb fully followed liver zonation and the Wnt/ β -catenin regulation, except for data of Supplemental Fig.9, that will be discussed below. I do understand that the values may be heterogeneous inside a same group, but I am surprised that the quantification revealed a so small SD for the means of all the genes.

2. Concerning the data related to the Wnt pathway.

a. The mention of the paper of Benhamouche et al, Dev Cell, 2007 should be added, as it is the first paper that described the role of the Wnt/ β -catenin signalling in liver zonation.

b. Data of Supplemental Fig.9 are still erroneous for Glul in the β -catenin-KO group, at least in the untreated group (See Data source file) and I don't see "how the mistake for Glul has corrected" as claimed by the authors in the reviewer response. Glul is a direct target of the Wnt/ β -catenin signalling and cannot be strongly up-regulated in the untreated β -catenin-KO compared to the WT as shown in Supplemental Fig.9 that correspond to data source file. This is a serious problem that should be resolved. Furthermore, if Glul is up-regulated in the 24h post-PH in β -catenin-KO as shown in Supplemental Fig.9, it should indicate that the functional compensation for Glul is β -catenin-independent, because it should be very low in the β -catenin-KO untreated group. This is consistent with data observed in EC-WIs (Fig.5e). The conclusion in the result section (lines 280-286) is thus not correct and should be deleted.

c. One of the main benefit to use β -catenin-KO together with EC-WIs and Mac-WIs is to try to define the role of the canonical and non-canonical Wnt pathways during the functional compensation.

Studies with the EC-WIs mice that mimics β -catenin-KO revealed a decrease in proliferation 24h post-PH that should correspond to the delayed regeneration already described in previous works (Sekine et al, Hepatology 2007; Torre et al, J Hepatol 2011). Data with EC-WIs and β -catenin-KO mice seem to show that the functional compensation is independent from the Wnt/ β -catenin signalling for the positive β -catenin target genes (that is not unexpected), but also for genes not controlled by the β -catenin signalling. However, this later data stands on only one gene (Alb). Concerning Arg1, the functional compensation in β -catenin-KO after PH may be linked to the fact that it is a negative target of the β -catenin.

As pointed by the authors, data with the Mac-WIs mice showed that the involvement of the Wnts secreted by the macrophages for liver proliferation during regeneration after PH is less critical than Wnts secreted from endothelial cells. This is in agreement with work published by the authors and should be cited (Yang et al Hepatology 2014). These Wnts appear important for the functional compensation as noted by the authors.

Finally, all these data are based on only four genes in which three are controlled by β -catenin including two positive and one negative and only one that is not controlled by β -catenin. I do

understand that is a lot of work with cutting-edge technologies, and I do not request studies with the other genes described by the authors in Fig3. But the authors should be more careful in their conclusions and should integrate in their discussion the results of previous works on the role of the Wnt signalling in liver homeostasis and during liver injury.

Point-by-point response to “Functional Compensation Precedes Recovery of Tissue Mass Following Acute Liver Injury” by Walesky et al.

NB Reviewer comments are provided in bold; *responses in blue italics.*

Reviewer #1 (Remarks to the Author):

I appreciate the additional analysis the authors performed to clarify and bolster the point that non-cycling hepatocytes are more involved in functional compensation during regeneration. The additional findings of the different functions between endothelial-derived Wnts and macrophage-derived Wnts is quite interesting. I still have concerns that the authors were not able to maximize the utility of a single cell-based approach to further delineate heterogeneity within hepatocytes at each of the time points in the study. However, as a whole the authors have shown convincingly novel insights about functional compensation in hepatocytes in response to injury.

We thank the Reviewer for her/his supportive words, and helpful suggestions that have improved the quality of our experimental data and the impact of our study. We appreciate that there are still several outstanding questions, including some regarding heterogeneity among hepatocytes during the course of the injury response. We firmly believe that the expansive data set we have generated here can be used in future studies to further examine these and other liver biology questions.

Reviewer #2 (Remarks to the Author):

In the revised manuscript, the authors provided more details on their analyses and included additional smRNA-FISH data as well as experimental data on mouse Wntless knockout model. They have satisfactorily addressed all my concerns. My remaining suggestion is to increase the font size for the figures to improve readability.

We thank the Reviewer for her/his helpful comments and suggestions, which encouraged us to expand our existing datasets and improve the impact of our manuscript. As suggested, we have now increased the font size throughout all figures to enhance readability. We will work with the editorial and production staff of the journal to ensure that our figures meet publication standards.

Reviewer #3 (Remarks to the Author):

I clearly appreciate the improvement of the manuscript that makes the messages clearer. However, there are still surprising results with the β -catenin-KO model that impact on the conclusions raised by the authors. Furthermore, the discussion should include previous works from other groups and from their own groups on the role of the Wnt/ β -catenin signaling in liver homeostasis and during liver regeneration/injury.

We thank this Reviewer for the recommendations and suggestions to further improve our manuscript. We have now addressed the remaining concerns described below.

1. First, if I am correct I do not find the description of the method used by the authors to establish the average mRNA expression from the smFISH data. In the data source file, the SD for all the genes are

relatively small, although the different values between each three mice of a same group are rather heterogeneous. This is the case for most of the genes studied. I did not put in question the quality of the smFISH data that showed expected results for the different genes studied. Expression of Glul, Cyp2e1, Arg1, Alb fully followed liver zonation and the Wnt/ β -catenin regulation, except for data of Supplemental Fig.9, that will be discussed below. I do understand that the values may be heterogeneous inside a same group, but I am surprised that the quantification revealed a so small SD for the means of all the genes.

We appreciate the Reviewer's interest in our methods. For the experiments depicted in Figure 5, we wanted to highlight the general impact of Wnt signals derived from different sources on the overall transcriptional response across the entire lobule. While we have shown in prior figures in great detail the heterogeneity of transcriptional changes across the lobule in relation to the distance from the central vein, the focus here is to have a measure of the "Average Total RNA Expression" as represented by the AUC (area under the curve) for RNA transcripts detected by smFISH across the liver lobule. In that sense, the AUC represents an average measure of gene expression per lobule, which is largely consistent across different areas. Thus, the resulting average of several AUC measures has a small SD, and highlights the significant differences observed between endothelial and macrophage Wntless knockout. A description of the method used to establish the average mRNA expression has been added to the methods section to enhance clarity (line 802).

2. Concerning the data related to the Wnt pathway.

a. The mention of the paper of Benhamouche et al, Dev Cell, 2007 should be added, as it is the first paper that described the role of the Wnt/ β -catenin signalling in liver zonation.

We appreciate the comment to highlight this landmark paper on liver zonation, which is truly important with regard to APC and Wnt/ β -catenin regulation. This paper has now been referenced (line 249).

b. Data of Supplemental Fig.9 are still erroneous for Glul in the β -catenin-KO group, at least in the untreated group (See Data source file) and I don't see "how the mistake for Glul has corrected" as claimed by the authors in the reviewer response. Glul is a direct target of the Wnt/ β -catenin signalling and cannot be strongly up-regulated in the untreated β -catenin-KO compared to the WT as shown in Supplemental Fig.9 that correspond to data source file. This is a serious problem that should be resolved. Furthermore, if Glul is up-regulated in the 24h post-PH in β -catenin-KO as shown in Supplemental Fig.9, it should indicate that the functional compensation for Glul is β -catenin-independent, because it should be very low in the β -catenin-KO untreated group. This is consistent with data observed in EC-WIs (Fig.5e). The conclusion in the result section (lines 280-286) is thus not correct and should be deleted.

We would like to thank the Reviewer for her/his careful examination of the data. We have now repeated the Glul expression measurements both in wild-type and β -catenin knockout mice. To exclude any possibility of misidentifying autofluorescence within erythrocytes as contributing to a positive smFISH signal, we now analyzed these data with more stringent FISH-quant parameters. This revealed a decrease in overall Glul expression in the β -catenin KO mice, as expected by the Reviewer. We have examined all of our data to minimize the contribution of autofluorescence in any of the other data sets.

Importantly, loss of β -catenin severely diminished the compensatory transcriptional response. We have now modified the figure and corresponding text to reflect these results (line 280). Additionally, we have also modified the conclusion to highlight the potential role of non-canonical Wnt signaling as a

mechanism for functional compensation and included a statement that further studies will be needed for validation (line 340).

c. One of the main benefit to use β -catenin-KO together with EC-WIs and Mac-WIs is to try to define the role of the canonical and non-canonical Wnt pathways during the functional compensation. Studies with the EC-WIs mice that mimics β -catenin-KO revealed a decrease in proliferation 24h post-PH that should correspond to the delayed regeneration already described in previous works (Sekine et al, Hepatology 2007; Torre et al, J Hepatol 2011). Data with EC-WIs and β -catenin-KO mice seem to show that the functional compensation is independent from the Wnt/ β -catenin signalling for the positive β -catenin target genes (that is not unexpected), but also for genes not controlled by the β -catenin signalling. However, this later data stands on only one gene (Alb). Concerning Arg1, the functional compensation in β -catenin-KO after PH may be linked to the fact that it is a negative target of the β -catenin.

We appreciate the need to put our results in the context of prior work. The papers mentioned above have now been referenced in the manuscript (line 320) to point to the role of endothelial Wnts for the proliferative response. The conclusion for this section has also been modified to include the possibility of a non-canonical (β -catenin-independent) mechanism to regulate the functional compensation discovered here (line 280).

As pointed by the authors, data with the Mac-WIs mice showed that the involvement of the Wnts secreted by the macrophages for liver proliferation during regeneration after PH is less critical than Wnts secreted from endothelial cells. This is in agreement with work published by the authors and should be cited (Yang et al Hepatology 2014). These Wnts appear important for the functional compensation as noted by the authors.

We appreciate the Reviewer's suggestion to highlight the previous work regarding the role of endothelially secreted Wnts for the proliferative response during liver regeneration. The paper by Yang et al. has now been referenced (line 331).

Finally, all these data are based on only four genes in which three are controlled by β -catenin including two positive and one negative and only one that is not controlled by β -catenin. I do understand that is a lot of work with cutting-edge technologies, and I do not request studies with the other genes described by the authors in Fig3. But the authors should be more careful in their conclusions and should integrate in their discussion the results of previous works on the role of the Wnt signalling in liver homeostasis and during liver injury.

We recognize the Reviewer's concern regarding the limitations of our study with regard to the number of genes confirmed by sm-FISH. The conclusions referenced in this section have now been modified to include additional potential mechanisms, including a role for non-canonical Wnt signaling. We have also added all of the above referenced papers to further highlight previous work on the role of Wnt signaling in liver homeostasis and injury.

REVIEWERS' COMMENTS:

Reviewer #3 (Remarks to the Author):

The authors have correctly answered to my concerns.